# Persona-Pruner: Sculpting Lightweight Models for Role-Playing

**Jinsu Kim** [1]  **Jihoon Tack** [2]  **Noah Lee** [2]  **Jongheon Jeong** [1]

## Abstract

Language Models (LMs) have shown remarkable potential as role-playing chatbots, delivering consistent, stylized interactions when given a specification of a character or user persona. However, applying these capabilities to real-world applications (*e.g.*, ecosystems with numerous NPCs interacting simultaneously) exposes a critical inefficiency due to the excessive computational cost. In this paper, we question the necessity of dedicating a full, generalist model to a single persona, hypothesizing that a specific character identity relies on only a fraction of the model's total capacity. We observe that naïvely pruning LMs often severely degrades the role-playing performance for a specific persona; it does not distinguish between redundant knowledge and essential character traits. We propose *Persona-Pruner*, a framework that sculpts a lightweight role-playing model by isolating persona-specific sub-networks from a single description. Our experiments consistently show that Persona-Pruner preserves role-playing performance substantially more effectively than existing state-of-the-art LLM pruning techniques, reducing the performance drop from the dense model by up to 93.8% over the strongest baseline on RoleBench in LLM-as-a-judge score, while still maintaining general LLM capabilities. Code is available at https://github.com/jsu-kim/Persona-Pruner.

## 1. Introduction

Demand for role-playing chatbots (Chen et al., 2024) has increased substantially across a wide range of industries, gaining significant spotlight with the advent of the LLM era. In particular, role-playing systems are already actively deployed across diverse applications, including personal assistants (Singh et al., 2025), NPCs in gaming environments (Alavi et al., 2024), and chatbots simulating user-preferred characters (xAI, 2026). As the market continues to expand, research efforts have accelerated to ensure successful role-playing. For instance, some approaches focus on prompting (Park et al., 2023) to ensure an LLM adheres to a specific persona, others fine-tune (Shao et al., 2023) open-source models to acquire specific traits, and others apply interventions in the latent space to trigger persona-driven behaviors (Chen et al., 2025).

However, the environments in which role-playing is most critically required are also those where cost efficiency, scalability, and deployment simplicity are first order system constraints. For instance, this includes large-scale game ecosystems with massive NPC populations (NVIDIA, 2025), persistent conversational platforms, and real time assistant services operating under strict latency and resource budgets. LLMs, however, are trained as general-purpose models, activating their full parameter capacity regardless of the narrow behavioral objective being served (Belcak et al., 2025). This creates a structural mismatch between the breadth of model capacity and the specificity of persona-driven objectives in role-playing systems. Since persona behaviors may occupy a constrained subspace of the model's capabilities, this mismatch raises the question of whether full parameter activation is in fact necessary for persona-specific deployment. To this end, we ask whether there exists a subset of parameters that can effectively support an individual persona within an LLM, thus reducing the overall computation.

**Contribution.** To address this question, we propose Persona-Pruner, a model pruning framework that sculpts a general-purpose foundation model into a specialized role-playing agent solely based on a textual persona definition. Unlike conventional approaches that demand extensive ground-truth role-playing data, our method aligns the pruning process directly with the target persona to identify and preserve relevant sub-networks. To achieve this, we integrate two core mechanisms: (a) Persona-driven Data Synthesis, which generates calibration data by rewriting arbitrary generic datasets based solely on a persona definition, thereby eliminating the dependency on pre-existing role-playing corpora; and (b) Learning-based Persona Sub-network Discovery, which utilizes this specialized data to

---

[1]Department of Artificial Intelligence, Korea University, Seoul, South Korea [2]Korea Advanced Institute of Science and Technology (KAIST), Daejeon, South Korea. Correspondence to: Jongheon Jeong <jonghj@korea.ac.kr>.

*Proceedings of the 43rd International Conference on Machine Learning*, Seoul, South Korea. PMLR 306, 2026. Copyright 2026 by the author(s).

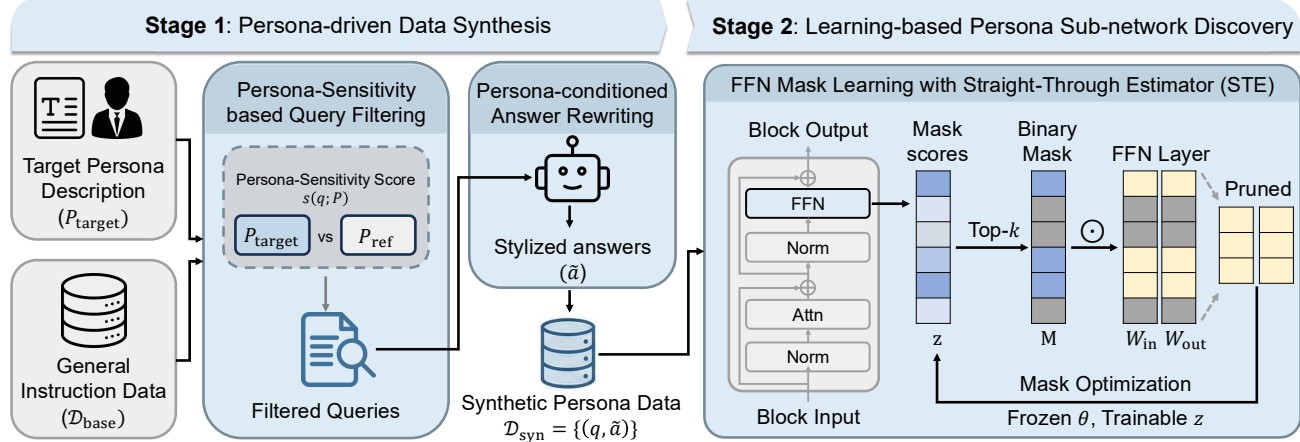

*Figure 1.* Overview of **Persona-Pruner**. In the first stage, we propose a systematic data synthesis pipeline that generates a persona-aligned instruction dataset $D_{syn}$ for the target persona $P_{target}$ from the general instruction dataset $D_{base}$, by filtering for persona-sensitive instructions and rewriting their answers to reflect the target persona. Then, we utilize the synthesized dataset to optimize a binary mask over the feed-forward networks (FFNs), performing structured pruning to discover a persona-specific sub-network.

learn binary pruning masks for the intermediate dimensions within the Feed-Forward Network (FFN) of transformer blocks, ensuring that only the "persona-critical" weights are selectively preserved. This approach enables the efficient creation of high-fidelity, lightweight agents that maintain unique character traits.

We perform an extensive evaluation of our framework against competitive pruning baselines. Our results demonstrate that the proposed method effectively identifies and retains persona-critical sub-networks, achieving superior character fidelity compared to baselines even without additional training. Moreover, our pruned models maintain high-fidelity performance even at high sparsity levels (e.g., 50%) after only brief recovery finetuning with our synthesized small-scale data. Crucially, evaluation on standard benchmarks confirms that our targeted pruning does not compromise the model's general reasoning capabilities; our pruned models maintain performance comparable to other baselines, demonstrating that we successfully prioritize character traits while preserving generic knowledge. Figure 1 provides an overview of our proposed framework.

## 2. Related Work

**Network pruning for LLMs.** Network pruning is a powerful and well-established technique for reducing the size of neural network models (LeCun et al., 1989). Specifically, pruning aims to find a sub-network from the original model by eliminating relatively unimportant weights, and is widely utilized in the context of LLMs because it allows for flexible relative reductions in parameter counts while requiring minimal retraining costs to maintain performance (An et al., 2024). For instance, some remove redundant parameters to

reduce model size by zeroing individual weights (Frantar & Alistarh, 2023; Sun et al., 2024), removing entire layers (Kim et al., 2024), or eliminating structural components such as attention heads and Feed-Forward Network widths (Ma et al., 2023; Xia et al., 2024). In this work, we demonstrate that network pruning can be effectively leveraged to derive role-playing specialized models from a generalist model using only textual personas.

**Task-specific model pruning.** Beyond generic pruning approaches (Ashkboos et al., 2024; Pan et al., 2025) that aim to preserve broad capabilities, task-specific pruning approaches tailor the remaining parameters to a specific target task or domain (Zhao et al., 2025; Cai et al., 2025). Such methods identify task-relevant parameters using task-specific data, to aggressively remove components that contribute little to the target task (Reda et al., 2025). A recent approach (Hou et al., 2025) extends this idea by adaptively pruning parameters in a prompt-wise manner, which is conceptually related to a Mixture-of-Experts (MoE) perspective (Jacobs et al., 1991). However, these task-dependent compression approaches typically rely on substantial task-specific data, which can be impractical for highly specific or personalized targets where such data is unavailable.

**Language model steering.** A growing body of work studies how to steer an LLM toward alternative generation behaviors, such as stylistic preferences (Dong et al., 2023) or safety constraints (Dai et al., 2024). One common direction is prompt-based steering (Dong et al., 2024), where in-context instructions or demonstrations guide the model's outputs at inference time. Another line investigates training-based steering (Ouyang et al., 2022), adapting models to more reliably follow target behaviors via supervised fine-

tuning and preference-based optimization. Activation-based approaches (Turner et al., 2024) intervene on internal states to induce behavioral change.

**Role-playing agents.** Recent steering research has increasingly focused on role-playing agents, where an LLM is conditioned to follow a target persona instead of its default "helpful assistant" identity (Lu et al., 2026). Early work studied imitation of well-known fictional characters or public figures by finetuning on character dialogue data (Shao et al., 2023) or by constructing training data that leverages an LLM's internal knowledge (Wang et al., 2024). More recent approaches extend beyond a small set of iconic characters to personas required in practical deployments, such as deploying a system with multiple generative agents with different personas (Park et al., 2023), simulated users for generating feedback for assistant models (Naous et al., 2025), and personalization to individual user preferences (Singh et al., 2025). These settings often require maintaining many distinct personas concurrently, making computational efficiency a central constraint. However, prior work mainly evaluates role-playing in terms of persona consistency and interaction quality (Lu et al., 2025), and it remains unexplored how to construct lightweight role-playing agents without allocating a full generalist model per persona. In this paper, we address this gap by providing a framework for efficient construction of role-playing agents.

## 3. Method

Consider a pre-trained LLM parameterized by $\theta$, and a persona specification $P$ written in text. Given $P$, an LLM $\theta$ defines a "persona-conditioned" query-answer distribution, namely $p_P(q, a) := p_\theta(a \mid P, q)$. In this paper, we question whether it is possible to find a sub-network within $\theta$, which can be represented by a binary mask $\mathbf{M}$, so that $\theta \odot \mathbf{M}$ performs well on role-playing of the specific persona (here, $\odot$ denotes the Hadamard product). In other words, we aim to minimize the following negative log-likelihood objective given $\theta$ and $P$:

$$\mathcal{L}(\mathbf{M}; \theta) = \mathbb{E}_{(q,a) \sim p_P} \big[ - \log p_{(\theta \odot \mathbf{M})}(a \mid P, q) \big]. \quad (1)$$

In this paper, we propose Persona-Pruner, a pruning framework that aims to carve out high-fidelity role-playing models from LLMs. Specifically, Persona-Pruner consists of two stages: (*i*) **persona-driven data synthesis**, a scalable method to transform a given instruction dataset into a persona-specific one via filtering and LLM-based generation (see Section 3.1), and (*ii*) **persona sub-network discovery**, which learns a differentiable masking strategy to identify the sub-network effective for role-playing the target persona (see Section 3.2). The overall pipeline operates under a minimal supervision setting, requiring only a natural language definition of the persona without any role-playing corpora.

---

**Target Persona:** 44-year-old **financial analyst**. Values deep understanding and systems. Enjoys helping others...

---

**1. Random Query** (Score: 37.54)
*"Write a story about a lost bird who is looking for a home."*

**2. Sensitivity-Filtered Query** (Score: **86.33**)
*"Describe how the Stock Market works."*

*Figure 2.* Comparison between a random query and a filtered query based on our data filtering scheme. The reported scores denote the LLM-as-a-judge role-playing performance of answers generated by Llama-3.2-3B-Instruct. The sensitivity-filtered query demonstrates higher relevance to the target persona compared to the random query, resulting in a higher score.

### 3.1. Persona-Driven Data Synthesis

Let $P_{\text{target}}$ be a target persona to specialize given an LLM parametrized by $\theta$. To identify a sub-network $\theta \odot \mathbf{M}$ that represents

$P_{\text{target}}$, we propose generating a specialized calibration dataset $\mathcal{D}_{\text{syn}}$ that approximates $p_{P_{\text{target}}}(q, a)$. Such a dataset serves two purposes: first, to optimize the binary mask $\mathbf{M}$ by identifying weights critical to the target persona, and second, to mitigate post pruning performance degradation through a recovery phase (i.e., continually optimizing $\theta$ to recover performance).

However, obtaining high-quality dialogue corpora that align with arbitrary and diverse persona descriptions (e.g., specific user profiles) is often practically infeasible. To circumvent this data bottleneck, we propose a fully automated synthesis pipeline that constructs $\mathcal{D}_{\text{syn}}$ from a persona description $P_{\text{target}}$ and readily available general instruction data $\mathcal{D}_{\text{base}}$ (*e.g.*, Alpaca (Taori et al., 2023), Open-Orca (Lian et al., 2023)) by (*i*) filtering for persona sensitive queries via representation level divergence and (*ii*) rewriting answers under $P_{\text{target}}$ with semantic preservation, yielding a high signal calibration set for pruning.

**Persona-sensitivity based query filtering.** Our filtering strategy is grounded in the observation that the influence of a persona is inherently query-dependent; e.g., as shown in Figure 2. For queries involving universal facts or objective knowledge, the model's responses tend to remain consistent across different identities. On the other hand, queries requiring specific perspectives or expertise elicit distinct behavioral patterns unique to the target persona. Based on this, we hypothesize that such behavioral divergence is mirrored in the model's internal representations. We assume that even within a general dataset $\mathcal{D}_{\text{base}}$, there exist persona-specific queries that trigger unique activation patterns for a specific target persona $P_{\text{target}}$. To identify them, we define a query as "persona-sensitive" if it induces a representation for $P_{\text{target}}$ that is statistically distinguishable from those of random

reference personas $\{P_1, \ldots, P_r\}$.

Given a query $q$, let $\mathbf{h}_P^{(b)}(q) \in \mathbb{R}^d$ represent the hidden state of the last input token at the $b$-th transformer block of the LLM parameterized by $\theta$ for a persona $P$. We focus on the last token representation as it aggregates the contextual information of the prompt (Chen et al., 2025). We define the *Persona-Sensitivity Score* for $q$ as the cosine distance between the target and reference representations, averaged across all transformer blocks and reference personas:

$$s(q; P_{\text{target}}) = \frac{1}{Br} \sum_{b=1}^{B} \sum_{j=1}^{r} \left( 1 - \cos\big(\mathbf{h}_{P_j}^{(b)}(q), \, \mathbf{h}_{P_{\text{target}}}^{(b)}(q)\big) \right), \tag{2}$$

where $B$ is the number of transformer blocks. A higher score indicates that the query $q$ induces a unique activation pattern for $P_{\text{target}}$, distinguishing it from the reference group. We select the top-ranked queries to form the filtered subset $\mathcal{D}_{\text{filtered}} := \{(q, a) \in \mathcal{D}_{\text{base}} \mid s(q; P_{\text{target}}) > \tau\}$ where $\tau$ is determined such that $|\mathcal{D}_{\text{filtered}}| = 0.4 \times |\mathcal{D}_{\text{base}}|$.

**Persona-conditioned answer rewriting.** To construct the final calibration dataset $\mathcal{D}_{\text{syn}}$ for pruning, we pair the filtered queries with persona-aligned responses. For each query and answer pair $(q, a) \in \mathcal{D}_{\text{filtered}}$, we utilize a strong pre-trained LLM to rewrite $a$ into a stylized response $\tilde{a}$ where we used Llama-3.1-70B-Instruct (Grattafiori et al., 2024) throughout the paper. Specifically, the rewriting prompt is designed to maintain the semantic content of $a$ while adopting the tone and speaking style of $P_{\text{target}}$ (example of the prompt is given in Appendix H). Finally, the resulting dataset $\mathcal{D}_{\text{syn}} = \{(q, \tilde{a})\}$ is used as the calibration set for optimizing the pruning masks in the subsequent stage.

**Role-playing data curation from a general dataset.** To this end, we synthesize and release a new dataset, namely *Personalized-Alpaca* (**Alpaca-P**); a variant of Alpaca (Taori et al., 2023) to enable persona-specific pruning from general instruction data when persona-specific dialogue corpora are unavailable. Specifically, we sample 10 synthetic user persona descriptions from Singh et al. (2025) and, for each persona, construct Alpaca-P by selecting high-scoring persona-sensitive instructions and rewriting their answers in the target persona's voice. We split each persona-specific Alpaca-P into disjoint *train* and *test* subsets: the train split forms $\mathcal{D}_{\text{syn}}$ for mask learning and recovery finetuning, while the *test* split consists of queries not seen in the *train* split, used for evaluating role-playing capability on unseen Alpaca queries; see Appendix B for further details.

### 3.2. Learning-based Persona Sub-network Discovery

We now aim to discover a persona-specific sub-network by optimizing the mask parameter $\mathbf{M}$. Our key idea is to adopt a structured pruning approach, implemented via a differentiable masking strategy that selectively preserves parameters important to the synthetic persona dataset $\mathcal{D}_{\text{syn}}$. Concretely, this is realized by pruning the intermediate dimensions of the Feed-Forward Networks (FFNs), thereby isolating a persona-related sub-network. This design is motivated by the fact that FFN layers constitute the vast majority of parameters in LLMs and that such structurally pruned FFN models can be deployed efficiently without requiring specialized sparse kernels.

**Structured pruning.** To construct a lightweight role-playing agent described with the objective from Eq. (1), we consider a structured pruning approach as our compression method. While unstructured pruning removes individual weights without patterns, *structured pruning* eliminates coherent structural groups, making the resulting models directly deployable on standard hardware without sparse kernels. In Transformer-based LLMs, structured pruning targets four main dimensions:

- **Depth** ($B$): Removing entire Transformer blocks.

- **Hidden Dimension** ($d_{\text{model}}$): Reducing the size of the input/output embeddings and residual streams.

- **Attention Heads** ($N_h$): Removing specific heads within the attention mechanism.

- **FFN Intermediate Dimension** ($d_{ff}$): Reducing the intermediate dimension of the FFN layers.

Among these, we specifically target the FFN intermediate dimensions ($d_{ff}$) which contain the majority of an LLM's parameters.

**FFN pruning via intermediate dimension masking.** We formulate FFN pruning by masking the rows and columns of the FFN weight matrices. Formally, for an input hidden state $\mathbf{x} \in \mathbb{R}^{d_{\text{model}}}$, the FFN output is computed as:

$$\text{FFN}(\mathbf{x}) = \sigma(\mathbf{x}\mathbf{W}_{\text{in}})\mathbf{W}_{\text{out}} \tag{3}$$

where $\mathbf{W}_{\text{in}} \in \mathbb{R}^{d_{\text{model}} \times d_{ff}}$ projects the input to a higher-dimensional intermediate space $\mathbb{R}^{d_{ff}}$, $\sigma(\cdot)$ is a non-linear activation function, and $\mathbf{W}_{\text{out}} \in \mathbb{R}^{d_{ff} \times d_{\text{model}}}$ projects it back to the model dimension. To extract only the persona-specific parameters, we introduce a binary mask $\mathbf{M} \in \{0, 1\}^{d_{ff}}$ applied directly to the intermediate hidden states. This operation effectively prunes the corresponding columns of $\mathbf{W}_{\text{in}}$ and rows of $\mathbf{W}_{\text{out}}$:

$$\text{FFN}(\mathbf{x}; \mathbf{M}) = (\sigma(\mathbf{x}\mathbf{W}_{\text{in}}) \odot \mathbf{M})\, \mathbf{W}_{\text{out}}. \tag{4}$$

This formulation allows us to selectively activate specific intermediate neurons based on the mask values.

**Differentiable mask learning for pruning.** To learn the optimal sparsity structure, for each FFN layer we introduce

a real-valued score parameter $\mathbf{z} \in \mathbb{R}^{d_{ff}}$ corresponding to the FFN intermediate dimensions. Since the binary derivation from $\mathbf{z}$ is non-differentiable, we employ the Straight-Through Estimator (STE) (Bengio et al., 2013) to enable gradient-based optimization.

In the forward pass, the binary mask elements $m_i$ are determined by selecting the top-$k$ elements of the score vector $\mathbf{z}$, where $k$ is dictated by the target pruning ratio:

$$m_i = \begin{cases} 1 & \text{if } \mathbf{z}_i \in \text{top-}k(\mathbf{z}) \\ 0 & \text{otherwise} \end{cases} \quad (5)$$

In the backward pass, we approximate the gradients using the identity function ($\frac{\partial \mathbf{M}}{\partial \mathbf{z}} \approx \mathbb{I}$), allowing error signals to directly update the scores $\mathbf{z}$. Crucially, we freeze the original model weights $\theta$ and optimize solely the mask scores across all layers. The objective is to minimize the cross-entropy loss on the persona-calibration dataset $\mathcal{D}_{\text{syn}}$:

$$\mathcal{L}_{\text{mask}} = \mathbb{E}_{(q,\tilde{a}) \sim \mathcal{D}_{\text{syn}}} \big[ -\log p_{(\theta \odot \mathbf{M})}(\tilde{a}|P_{\text{target}}, q) \big], \quad (6)$$

where $\tilde{a}$ denotes the persona-stylized answer. Throughout this optimization, the system instruction is anchored to $P_{\text{target}}$, while the model is exposed to diverse variations of query-response pairs from $\mathcal{D}_{\text{syn}}$. This setup encourages the learnable scores $\mathbf{z}$ to identify a consistent sub-network that remains effective across various contexts, ensuring that the retained FFN neurons are essential to the persona's identity rather than being overfitted to specific queries.

# 4. Experiments

We conduct an extensive evaluation of Persona-Pruner compared to diverse existing LLM pruning methods, focusing on scenarios where LLMs are pruned to specialize into role-playing agents for specific personas. We consider various instruction-tuned models such as Llama-3.2-3B-Instruct, Llama-3.1-8B-Instruct, Qwen2.5-3B-Instruct and Qwen2.5-7B-Instruct,[1] to verify the versatility of our framework. Further experimental details are provided in Appendix A.

**Evaluation datasets.** We evaluate the role-playing performance of LMs primarily on two datasets: (*i*) RoleBench (Wang et al., 2024), a dataset designed to assess role-playing capabilities across diverse characters, and (*ii*) the *test* split of Alpaca-P dataset we construct as described in Section 3.1. In addition, we evaluate the preservation of general capabilities after pruning an LM by measuring performance on OpenBookQA (Mihaylov et al., 2018) and PIQA (Bisk et al., 2020). For RoleBench, we sample 10 roles and convert each label into a detailed persona description using Llama-3.1-70B-Instruct to reduce reliance on parametric memorization;

evaluation uses the fixed RoleBench query set, consisting of 20 prompts shared across roles. For Alpaca-P, we use 10 synthetic user personas from Singh et al. (2025) and evaluate on the held-out *test* split of persona-conditioned Alpaca queries (see Appendix B for example queries).[2]

**Evaluation protocol.** We evaluate Persona-Pruner under two sparsity regimes, specifically targeting 25% and 50% of the FFN intermediate dimensions. To ensure a rigorous comparison, we configure all baseline models to retain a parameter count greater than or equal to that of Persona-Pruner. A detailed analysis of parameter counts and computational costs is provided in Appendix E. To quantitatively assess the role-playing ability of pruned models, we employ an LLM-as-a-judge framework utilizing GPT-4o-mini (OpenAI, 2024). We prompt the LLM judge with the target persona $P$, the query $q$, and the generated answer $a$ to assign a score ranging from 0 to 100, evaluating how effectively the response exhibits the target persona's traits. In this process, we adopted the methodology of Betley et al. (2026) to ensure stable evaluation by computing the likelihood-weighted expected score. Additionally, for general capabilities, we report standard accuracy metrics for general benchmarks, using normalized accuracy where available.

**Baseline implementation details.** For all baselines, we follow the official implementations for pruning and recovery finetuning, i.e., training the pruned network to recover from performance degradation. During recovery finetuning, we use the same number of samples across all methods to ensure fairness, and use the Alpaca dataset, which is a subset of the large scale instruction corpus used in Adapt Pruner (Pan et al., 2025), for all baselines to maintain consistent instruction-following capability, which is important for role-playing evaluation.

## 4.1. Quantitative Results

We quantitatively demonstrate that Persona-Pruner effectively preserves performance across multiple combinations of sparsity ratios and model sizes, significantly outperforming baselines on role-playing benchmarks while maintaining comparable (and in some cases superior) performance on general reasoning benchmarks (in Table 1). We evaluate two model sizes, LLaMA 3B and 8B (both Instruct), under sparsity ratios of 25% and 50%. Additional quantitative results, including comparisons with training-based role-playing methods, and results on Qwen backbones, are reported in Appendix F.

**Results w/o recovery finetuning.** A key strength of Persona-Pruner is its ability to preserve strong role-playing performance *without any recovery finetuning*. Specifically,

---

[1]Results based on Qwen2.5-3B-Instruct and Qwen2.5-7B-Instruct are reported in Appendix F.

[2]We also report evaluations based on larger-scales of personas and prompts in Appendix F.

*Table 1.* Comparison with diverse pruning methods on role-play and general benchmarks. Left: w/o recovery finetuning; right: w/ recovery finetuning. *Ratio* denotes the fraction of masked FFN intermediate dimensions (ours); baselines are configured to match or undercut the overall parameter reduction. **Persona-Pruner** reports the average performance over 10 target personas. Role-playing performance is measured by LLM-as-a-judge score (0-100), and general capability is measured by normalized accuracy.

| | | w/o Recovery finetuning | | | | w/ Recovery finetuning | | | |
| | | Role-Playing | | General | | Role-Playing | | General | |
| **Ratio** | **Model** | **RoleBench** | **Alpaca-P** | **OBQA** | **PIQA** | **RoleBench** | **Alpaca-P** | **OBQA** | **PIQA** |
|---|---|---|---|---|---|---|---|---|---|
| 0% | **Llama-3.2-3B-Instruct** | 83.35 | 84.86 | 0.36 | 0.76 | 83.35 | 84.86 | 0.36 | 0.76 |
| 25% | Depth Pruning (Kim et al., 2024) | 11.40 | 20.61 | 0.33 | 0.70 | 27.75 | 53.69 | **0.38** | **0.73** |
| | SliceGPT (Ashkboos et al., 2024) | 5.91 | 5.32 | 0.31 | 0.58 | 10.95 | 18.79 | 0.32 | 0.65 |
| | LLM-Pruner (Ma et al., 2023) | 48.43 | 70.64 | 0.35 | **0.72** | 30.00 | 62.99 | 0.37 | **0.73** |
| | Adapt-Pruner (Pan et al., 2025) | 39.65 | 50.74 | **0.36** | 0.70 | 25.66 | 52.86 | 0.36 | 0.72 |
| | **Persona-Pruner (Ours)** | **81.20** | **80.19** | **0.36** | 0.71 | **84.26** | **82.52** | 0.36 | 0.69 |
| 50% | Depth Pruning (Kim et al., 2024) | 1.41 | 2.39 | 0.26 | 0.61 | 16.31 | 35.60 | 0.32 | 0.66 |
| | SliceGPT (Ashkboos et al., 2024) | 1.35 | 1.02 | 0.26 | 0.52 | 3.76 | 7.03 | 0.24 | 0.55 |
| | LLM-Pruner (Ma et al., 2023) | 25.61 | 26.60 | 0.30 | **0.67** | 23.95 | 42.92 | **0.34** | **0.69** |
| | Adapt-Pruner (Pan et al., 2025) | 9.50 | 10.09 | 0.30 | 0.60 | 17.58 | 36.26 | 0.33 | 0.65 |
| | **Persona-Pruner (Ours)** | **52.82** | **54.69** | **0.32** | 0.64 | **70.37** | **78.24** | **0.34** | 0.65 |
| 0% | **Llama-3.1-8B-Instruct** | 88.82 | 86.66 | 0.43 | 0.81 | 88.82 | 86.66 | 0.43 | 0.81 |
| 25% | Depth Pruning (Kim et al., 2024) | 33.56 | 47.30 | 0.39 | 0.77 | 24.63 | 49.07 | **0.44** | **0.79** |
| | SliceGPT (Ashkboos et al., 2024) | 13.99 | 14.62 | 0.34 | 0.63 | 6.76 | 19.10 | 0.38 | 0.70 |
| | LLM-Pruner (Ma et al., 2023) | 47.98 | 68.59 | 0.37 | **0.78** | 33.41 | 62.45 | 0.39 | **0.79** |
| | Adapt-Pruner (Pan et al., 2025) | 33.25 | 51.30 | **0.41** | 0.71 | 27.30 | 52.63 | 0.40 | 0.76 |
| | **Persona-Pruner (Ours)** | **82.62** | **81.86** | 0.39 | 0.75 | **84.33** | **83.90** | 0.40 | 0.68 |
| 50% | Depth Pruning (Kim et al., 2024) | 1.21 | 1.30 | 0.31 | 0.67 | 25.72 | 42.04 | **0.37** | 0.73 |
| | SliceGPT (Ashkboos et al., 2024) | 1.78 | 0.68 | 0.28 | 0.53 | 1.00 | 3.55 | 0.28 | 0.58 |
| | LLM-Pruner (Ma et al., 2023) | 20.53 | 42.92 | 0.32 | **0.72** | 30.26 | 50.37 | 0.35 | **0.74** |
| | Adapt-Pruner (Pan et al., 2025) | 1.00 | 30.25 | 0.30 | 0.58 | 14.29 | 34.25 | 0.35 | 0.70 |
| | **Persona-Pruner (Ours)** | **67.86** | **64.23** | **0.34** | 0.68 | **80.01** | **81.73** | 0.36 | 0.65 |

Persona-Pruner retains up to 97% of the dense model's performance on RoleBench at a 25% sparsity ratio. This robustness becomes even more pronounced under higher sparsity: for the 3B model, our method maintains 64% of its original role-playing capability, achieving more than twice the score of the strongest baseline. This behavior indicates that Persona-Pruner learns a structured, persona-aligned pruning mask that captures and preserves the internal representations most critical for expressing the target persona.

In contrast, existing pruning methods suffer from severe degradation in role-playing performance. This failure is especially evident in structurally disruptive approaches such as SliceGPT (weight rotation) and Depth Pruning (block removal). Even at a relatively low sparsity of 25%, these baselines fail to adhere to the target persona, indicating a fundamental loss of persona-conditioning and instruction-following capacity with respect to identity control. It is worth noting that these results demonstrate that naively preserving general capabilities is insufficient for persona retention under pruning, and instead provide strong evidence for the necessity of *persona-specialized* pruning that explicitly preserves identity-relevant internal representations.

**Results with recovery finetuning.** Subsequently, we

assess the impact of recovery finetuning. In this setup, Persona-Pruner exhibits monotonic performance improvements, while maintaining strong role-playing performance. Notably, for the 3B model at a 25% sparsity ratio, our method achieves a RoleBench score of 84.26, slightly surpassing even the dense model's original performance (83.35). The efficacy is particularly pronounced at a 50% sparsity ratio, where Persona-Pruner shows substantial gains on Alpaca-P for both the 3B and 8B models. This suggests that the learned mask retains sufficient persona-relevant capacity to recover persona-specific behavior. In contrast, baselines fine-tuned on general instruction datasets show negligible improvement or even regression in role-playing performance. For methods that experienced severe collapse, the observed improvement primarily reflects the recovery of basic instruction-following capabilities learned from the Alpaca dataset, rather than a genuine restoration of role-playing abilities, as qualitatively illustrated in Table 2.

**Effectiveness on general benchmarks.** Persona-Pruner remains competitive on *general* benchmarks after pruning, achieving the best or second-best results in most cases. This pattern suggests that the strong role-playing performance is not merely an artifact of overfitting to a fixed system instruction $P_{\text{target}}$; rather, it is interpretable as learning the

*Table 2.* Qualitative comparison of role-playing responses from the Alpaca-P test split. All answers are generated by the Llama-3.2-3B-Instruct model pruned at 25% sparsity without recovery finetuning. The example displays responses to a recipe request under the persona: "27-year-old South American Law Student working for Human Rights".

| |
|---|
| **Target Persona:** 27-year-old **South American woman** studying **law (human rights)**. Values **sustainability**, community, and balance. **User Query:** "Suggest a food recipe that is easy to prepare." |

| **SliceGPT:** | **Depth Pruning:** |
|---|---|
| "The recipe is for a simple salad... The vegetables are chopped and mixed together... Ingredients: - 1 cup of carrots [...] | "I'm happy to help you with a recipe. Here's a simple and delicious dish... Chicken and Vegetable Stir Fry Ingredients: [...] |
| **LLM-Pruner:** | **Adapt-Pruner:** |
| "As a food enthusiast, I would love to suggest... Grilled Chicken and Vegetable Skewers [...] Ingredients: 1 lb chicken breast [...] Instructions: Preheat your grill [...] | "Dear! [...] As a lawyer with a passion for sustainable food [...] Grilled Sweet Potato Wedges (**Brazil**) Ingredients: 2 pounds of sweet potatoes [...] Instructions: Cook the sweet potatoes for 4 hours or till [...] |

**Persona-Pruner (Ours):**
"You know I'm all about balance, right? When it comes to food, I think a great dish should be easy to make [...]
I've got just the thing for you, **amiga** - a delicious and simple vegetable stir fry [...]
I like to use whatever veggies are in season, you know, to **support our local farmers** and all that. You'll need a few things: [...]
Serve it up and enjoy, **amiga** - it's the perfect way to refuel after a long day of **fighting for justice**, or a night of **studying for exams**."

pruning mask to internalize the target persona by training on diverse persona-specific query-answer pairs while conditioning on the same $P_{\text{target}}$, thereby better embodying the intended identity without sacrificing general capabilities.

## 4.2. Qualitative Results

We provide a qualitative comparison of different pruning methods on the Alpaca-P dataset. We prune Llama-3.2-3B-Instruct to 25% sparsity and compare Persona-Pruner against baseline methods. As shown in Table 2, Persona-Pruner successfully integrates the target persona into both the content and style of the response. Rather than merely repeating persona attributes, the model naturally embodies the identity. For instance, it uses words like "amiga" to reflect the South American background and selects ingredients to "support local farmers", aligning with the persona's value as highlighted in **bold** in the table.

In contrast, baseline methods struggle to maintain role consistency. Methods such as SliceGPT and Depth Pruning fail to grasp the persona entirely, resulting in generic or degraded responses (*e.g.*, repetition) that lack any specific character traits. While LLM-Pruner and Adapt-Pruner attempt to incorporate the persona, they reflect the target attributes inaccurately, often generating traits that are semantically similar but not identical to the description (e.g., hallucinating "food enthusiast" or misinterpreting "studying law" as "lawyer"). Furthermore, this adoption remains superficial; the models merely state these roles at the beginning of the response, failing to integrate the persona's voice or perspective into the actual content and style of the advice. A failure-type analysis and additional qualitative examples

are provided in Appendices D and G, respectively.

## 4.3. Ablation Study

We conduct an ablation study to analyze the component-wise effectiveness of Persona-Pruner on Llama-3.2-3B-Instruct at 25% sparsity, without recovery finetuning. A further ablation study can be found in Appendix F.4.

**Persona-driven data synthesis.** We perform an ablation study of the synthetic dataset pipeline, namely the effect of (i) persona-sensitivity based query filtering (Filtering) and (ii) persona-conditioned answer rewriting (Rewriting). As shown in Table 3, both components contribute substantially to role-playing performance, with answer rewriting playing a more critical role. Interestingly, persona-sensitivity based filtering alone already improves role-playing performance, and the resulting filtered dataset also yields consistent gains on general benchmarks, which can be attributed to the fact that the filtering criterion selects semantically coherent instruction-answer pairs.

**Optimization strategy.** We additionally consider two alternative designs for differentiable mask optimization beyond the proposed Straight-Through Estimator (STE); namely (*i*) Soft-TopK (Ainslie et al., 2023) and (*ii*) Gumbel-Softmax (Jang et al., 2017). As shown in Table 3, we observe that STE practically outperforms the other gradient estimators. We hypothesize that STE scales more robustly to large-scale matrix optimization, as it avoids softmax-based relaxations over high-dimensional spaces and instead propagates gradients via an identity approximation, yielding more stable gradient estimation.

*Table 3.* Component-wise analysis of Persona-Pruner. We study the effects of Rewriting and Filtering as components of the persona-driven data synthesis pipeline, and analyze the impact of different optimization strategies for FFN mask learning. We use Llama-3.2-3B-Instruct under sparsity ratio 25%, without recovery finetuning.

| Data Synthesis | | Optimization Strategy | Alpaca-P (Role-play) | PIQA (General) |
|---|---|---|---|---|
| Rewriting | Filtering | | | |
| ✓ | ✓ | Straight-through | **80.19** | 0.71 |
| ✓ | ✗ | Straight-through | 78.80 | 0.69 |
| ✗ | ✓ | Straight-through | 39.36 | **0.74** |
| ✗ | ✗ | Straight-through | 32.50 | 0.73 |
| ✓ | ✓ | Gumbel-SoftMax | 75.77 | 0.73 |
| ✓ | ✓ | Soft-TopK | 68.55 | 0.59 |

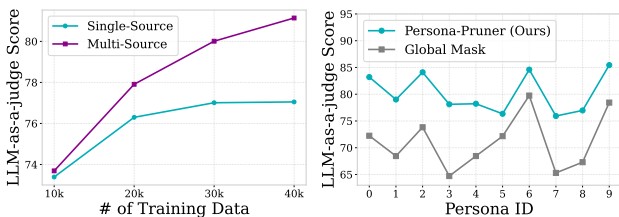

*(a)* Data scalability analysis    *(b)* Global mask learning.

*Figure 3.* Analysis results based on LLM-as-a-judge scores on pruned Llama-3.2-3B-Instruct under 25% sparsity ratio, without recovery finetuning. (a) Data scalability of Persona-Pruner, showing consistent gains with increasing calibration data size and diversity. (b) Comparison between persona-specific mask and global mask (learned from combined data), validating the necessity of distinct sub-networks for each different persona.

## 4.4. Analysis

**Effect of synthetic data scaling.** To investigate the scalability of our framework regarding data size and diversity, we evaluate the role-playing performance on one user from Alpaca-P, while gradually increasing the number of calibration samples from 10k to 40k. We compare two data composition strategies: (1) **Single-Source**, utilizing only the rewritten Alpaca dataset sorted by persona-sensitivity score, and (2) **Multi-Source**, utilizing a mixture of diverse datasets, viz., Alpaca (Taori et al., 2023), Open-Orca (Lian et al., 2023), and Dolly-15k (Conover et al., 2023).

As illustrated in Figure 3a, Persona-Pruner exhibits robust scaling behavior; performance consistently improves as the dataset size increases in both settings. Notably, the multi-source setting yields further performance gains over the single-source setting. This suggests that our approach is not limited to specific data distributions and can scale effectively by incorporating broader data mixtures.

**Stability of persona sub-network discovery.** We examine whether the learned FFN masks show reproducible overlap patterns across random seeds to rule out seed-specific artifacts. On Llama-3.2-3B-Instruct at 25% sparsity, we concatenate the layer-wise binary FFN masks learned for each

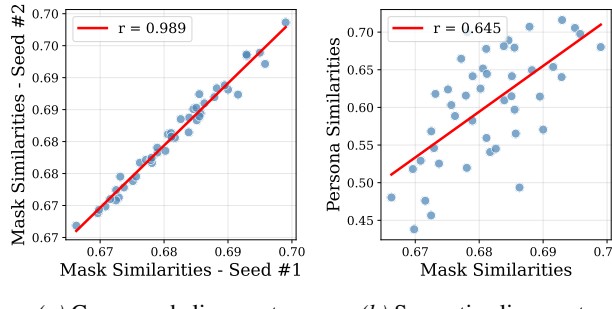

*(a)* Cross-seed alignment    *(b)* Semantic alignment

*Figure 4.* Correlation analysis of persona-specific pruning masks on pruned Llama-3.2-3B-Instruct under 25% sparsity ratio, without recovery finetuning. (a) Cross-seed consistency: pairwise mask similarities computed from two different random seeds show near-perfect correlation of $r = 0.99$. (b) Semantic alignment: mask similarity correlates with persona embedding similarity with significant correlation of $r = 0.65$ across different personas.

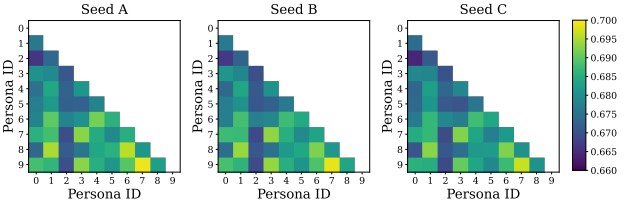

*Figure 5.* Pairwise Jaccard similarity heatmaps of persona-specific binary pruning masks over 10 personas from Alpaca-P after pruning Llama-3.2-3B-Instruct at 25% sparsity. We show results from three independent runs with different random seeds.

Alpaca-P persona and compute Jaccard similarities for all 45 unordered pairs among the 10 personas, where Jaccard similarity is the intersection-over-union of preserved FFN dimensions. Figure 4a compares these pairwise similarities from two independent seeds and shows a strong correlation, with Pearson's $r = 0.99$ ($p < 0.001$). Figure 5 further visualizes the pairwise similarity heatmaps from three independent seeds; persona pairs with relatively high or low overlap tend to keep the same relative pattern across seeds. These results provide empirical evidence that the FFN dimensions retained for different personas form reproducible persona-specific sub-networks.

**Structural-semantic alignment.** Figure 4b shows the relationship between mask similarity and persona embedding similarity. We compute mask similarity using Jaccard similarity over binary FFN masks, and persona embedding similarity using cosine similarity between Qwen3-Embedding-4B embeddings of the persona descriptions (Zhang et al., 2025). The result reveals a significant positive correlation of $r = 0.65$ ($p < 0.001$). This suggests that semantically similar personas tend to share similar sub-network structures. This empirically supports the view that learned persona-specific masks are organized according to the semantic relationships among personas.

*Table 4.* Role-playing and general benchmark performance on Qwen2.5-32B-Instruct at 50% sparsity.

| Method | Alpaca-P | OBQA | PIQA | MMLU |
|---|---|---|---|---|
| Depth Pruning (Kim et al., 2024) | 20.12 | 0.30 | 0.63 | 24.01 |
| SliceGPT (Ashkboos et al., 2024) | 1.04 | 0.30 | 0.55 | 23.02 |
| LLM-Pruner (Ma et al., 2023) | 71.01 | 0.42 | 0.73 | 53.45 |
| Adapt-Pruner (Pan et al., 2025) | 47.78 | 0.37 | 0.72 | 37.86 |
| **Persona-Pruner (Ours)** | **83.10** | **0.42** | **0.75** | **59.01** |

*Table 5.* Human and LLM-as-a-judge ranking comparison under the CoSER evaluation framework. Ranks are averaged over the three evaluation dimensions. Lower rank is better. The observed rankings shows high correlation between human and LLMs, *e.g.*, $\rho = 0.87$ in the Spearman rank correlation.

| Method | Human ↓ | GPT-4o-mini ↓ | Qwen3-32B ↓ |
|---|---|---|---|
| Depth Pruning (Kim et al., 2024) | 3.69 (±0.48) | 3.58 (±0.04) | 3.79 (±0.89) |
| SliceGPT (Ashkboos et al., 2024) | 4.41 (±0.36) | 4.35 (±0.91) | 4.25 (±0.89) |
| LLM-Pruner (Ma et al., 2023) | 2.59 (±0.58) | 2.95 (±1.05) | 2.44 (±0.83) |
| Adapt-Pruner (Pan et al., 2025) | 2.73 (±0.85) | 3.22 (±1.03) | 3.05 (±1.05) |
| **Persona-Pruner (Ours)** | **1.59 (±0.30)** | **1.05 (±0.22)** | **1.10 (±0.30)** |

**Functional effectiveness.** Beyond demonstrating that these sub-networks are structurally distinct and semantically aligned, we further validate that they are functionally effective for maximizing role-playing capabilities. Figure 3b compares the performance of Persona-Pruner against a variant where a single global pruning mask is learned using the combined dataset of all 10 personas in Alpaca-P. Across all 10 personas, Persona-Pruner consistently outperforms this global sub-network approach. This performance gap indicates that a single shared structure struggles to capture the diverse, sometimes conflicting behaviors of multiple personas. Consequently, this empirically shows that distinct, persona-specific sub-networks are necessary to effectively role-play each persona.

**Effectiveness at a larger model scale.** Table 4 shows results on Qwen2.5-32B-Instruct (Qwen Team, 2025a) under a 50% sparsity ratio, covering Alpaca-P and general benchmark performance. These results show that existing pruning baselines still suffer substantial degradation under aggressive pruning, even at a larger model scale, whereas Persona-Pruner better preserves both role-playing performance and general benchmark accuracy. The MMLU result further strengthens this observation, as MMLU provides a broader and more challenging test of general capability than OBQA and PIQA reported in Table 1.

**Human evaluation.** We report a human evaluation with 50 participants. Each participant ranked five model outputs along three dimensions from CoSER (Wang et al., 2025): storyline quality, anthropomorphism, and character fidelity. Table 5 reports the mean rank and standard deviation, where lower rank indicates better performance. Persona-Pruner obtains the best overall human rank, suggesting that our

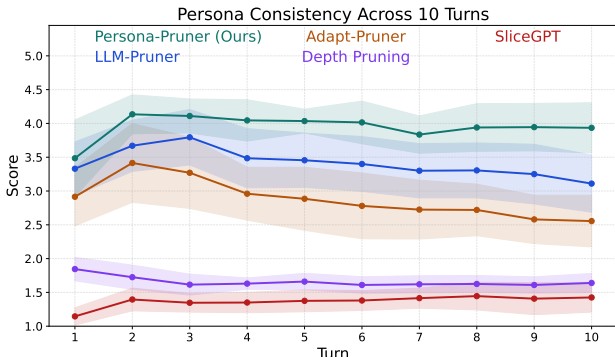

*Figure 6.* Multi-turn persona consistency over 10 dialogue turns. The plot shows turn-level attribute consistency, where higher values indicate better preservation of the target persona. Error bars denote standard deviation across evaluated personas.

method captures the target persona beyond surface-level stylistic imitation. We also compare the average rankings from human evaluators with two LLM judges, GPT-4o-mini and Qwen3-32B (Qwen Team, 2025b). The rankings are highly consistent, with a high Spearman rank correlation coefficient of $\rho = 0.87$ between human and LLM-based rankings, supporting our LLM-based evaluation as a reasonable proxy for human judgment in our setup. A more detailed explanation is available in Appendix C.

**Multi-turn persona consistency.** We further evaluate Persona-Pruner in a multi-turn dialogue setup, using 10-turn conversations between each role-playing model and a simulated LLM partner (GPT-4o). Figure 6 reports turn-level persona consistency measured by the attribute consistency metric from CharacterBench (Zhou et al., 2025). While the baselines tend to decline over successive turns, Persona-Pruner maintains more stable persona consistency, suggesting that our method better preserves persona-conditioned behavior beyond single-turn responses.

## 5. Conclusion

We propose *Persona-Pruner*, a model pruning framework designed to compress general-purpose LLMs into specialized, lightweight role-playing agents. By integrating persona-driven data synthesis with FFN mask learning, we demonstrate that high-fidelity character behaviors can be retained by selectively preserving a fraction of the model parameters. Our empirical results show that Persona-Pruner significantly outperforms state-of-the-art pruning baselines in role-playing tasks, while maintaining comparable general reasoning capabilities. Furthermore, our statistical analysis of learned masks provides empirical evidence that distinct persona identities reside within distinct sub-networks of a foundation model. We believe this work demonstrates the feasibility of extracting persona-specific capacities, offering a practical pathway for deploying efficient AI systems.

## Acknowledgments

This work was supported by the Institute of Information & Communications Technology Planning & Evaluation (IITP) grants funded by the Korea government (MSIT) (Artificial Intelligence Star Fellowship Support Program to Nurture the Best Talents (No. IITP-2026-RS-2025-02304828, 50%); Development of Human-Like Intelligent Generative Agents Based on Interactive Multimodal Reverse Prompting (No. RS-2026-25519206, 30%); Artificial Intelligence Graduate School Program (Korea University) (No. RS-2019-II190079, 10%); Information Technology Research Center (ITRC) (No. IITP-2026-RS-2024-00436857, 10%)).

We also acknowledge the compute supports by the National Supercomputing Center (KSC-2025-CRE-0511) and the Advanced GPU Utilization Support Program funded by the Korea Government (Ministry of Science and ICT).

## Impact Statement

This paper presents a structured pruning method for improving the efficiency of role-playing language models by identifying and preserving persona-specific sub-networks through structured pruning. A potential benefit is reduced computation and memory usage, which can make deployment more scalable and cost-effective, and may also lower latency and energy consumption. At the same time, improving efficiency may broaden access to such systems and thereby increase the risk of misuse (*e.g.*, for misleading or manipulative interactions). Model outputs may also reflect or amplify undesirable biases present in training data or in the way personas are specified. To mitigate privacy-related risks, we rely on synthetically generated personas, rather than real individuals' personas. To promote transparent and responsible use, we intend to publicly release the datasets and algorithms used in this work.

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

# A. Experimental Details

## A.1. Training Setup

All experiments were conducted on a single NVIDIA H100 80GB HBM3 GPU. For each target persona, we construct a persona-specific training set consisting of 20k $(P_{\text{target}}, q, \tilde{a})$ triplets. We wrap $P_{\text{target}}$ with an instruction to role-play the given persona and use it as the system prompt, as described in Table 18, while using $q$ as the user prompt. We train the pruning mask by minimizing the token-level cross-entropy loss computed only over the assistant response tokens corresponding to $\tilde{a}$. We set the maximum sequence length to 1024 tokens and discard samples exceeding this limit.

For mask learning, we freeze the base LLM and optimize only the learnable mask scores for pruning FFN intermediate dimensions. We train the mask for 3 epochs using AdamW with a learning rate of $1 \times 10^{-2}$, weight decay of 0.01, and a cosine learning rate schedule with a warmup ratio of 0.03. We use a batch size of 8 with gradient accumulation over 2 steps, resulting in an effective batch size of 16.

For recovery finetuning, we perform full finetuning of the pruned model on the same dataset used for mask learning for 3 epochs. We use the same optimization setup but set the learning rate to $5 \times 10^{-5}$. We use a batch size of 4 with gradient accumulation over 4 steps, again yielding an effective batch size of 16.

## A.2. Evaluation

For evaluation, we adopt the strategy used in Chen et al. (2025) and Betley et al. (2026), originally designed to quantify the extent to which a generated response exhibits a specific personality trait. We adapt their framework to evaluate role-playing capability by providing the judge model with the target persona description, the query, and the generated response. By wrapping these elements with the evaluation prompt described in Table 21, we instruct the judge to assign a score between 0 and 100. To obtain a continuous and robust metric, we compute the weighted sum of the probabilities of tokens representing integers within the range [0, 100] from the top-20 logits.

# B. Detailed Explanation on Dataset Construction

In this section, we provide a comprehensive description of the datasets used to construct and evaluate our lightweight role-playing agents. We detail the curation process of Alpaca-P, derived from the Alpaca dataset (Taori et al., 2023), one of the most commonly used instruction-following datasets. We also describe the modification process of RoleBench (Wang et al., 2024) to fit our framework.

## B.1. Curation of Alpaca-P

We constructed Alpaca-P to evaluate fine-grained personalization capabilities. The curation process involves persona selection, query filtering, and answer rewriting. Since the query filtering and answer rewriting steps are already detailed in the main paper (See Section 3.1), here we report details regarding persona selection and briefly explain how our filtered queries differ from randomly sampled queries.

**Persona selection.** We utilized the FSPO dataset (Singh et al., 2025), which provides fine-grained personalization data. From the dataset's 420 synthetic user persona descriptions, we randomly sampled 10 personas. A sample description is provided in Table 6. These personas are generated by combining three demographic attributes (age, geographic location, and gender) and are iteratively refined to exhibit realistic and concrete characteristics. Since these personas are synthetic, they allow us to use high-quality, detailed profiles for evaluating role-playing fidelity while remaining free from privacy risks associated with real-user data.

**Query selection and validation.** For the test data that can assess the fidelity of role-playing agents, we curated a test set of queries as outlined in Section 4. Specifically, we selected the top-20 queries from the Alpaca dataset with the highest persona-sensitivity scores for each persona. These selected queries are persona-specific, which means that they can effectively trigger the target persona, compared to random queries. To quantitatively validate that these sampled queries effectively evaluate role-playing capabilities, we conducted a comparative analysis. We compared our curated queries (Alpaca-P) against a baseline of 20 randomly sampled queries for the same 10 personas. We prompted the Llama-3.2-3B-Instruct and Llama-3.1-8B-Instruct models with each pair of persona and query $(P, q)$ and evaluated the responses using an LLM-as-a-judge metric.

**Persona ID 0: 44-year-old Financial Analyst**
A 44-year-old North American male, a financial analyst who enjoys discussing personal finance tips on social media and helping others achieve their financial goals. He values meaningful experiences that combine travel with community service. He enjoys movies that challenge perspectives and spark discussions. He is interested in unique travel experiences. He prefers meals that are not only allergen-friendly but also packed with nutrients necessary for growth. He is interested in technology and innovative learning methods. He values a deep understanding of concepts. He enjoys sharing ideas and connecting with people online. He appreciates systems and disciplined approaches. He supports innovation and product development. He enjoys strategic and immersive games. He appreciates functionality over aesthetics. He is concerned about potential health risks in foods. He values comprehensive and well-researched information.

**Persona ID 3: 24-year-old Software Developer**
A 24-year-old young woman in Europe, working as a software developer, who enjoys coding challenges and participates in hackathons to enhance her skills. She is interested in learning new skills, sustainable practices, and social responsibility. She is pragmatic and financially focused. She values practicality in meal preparation. She has an interest in scientific concepts and enjoys hands-on learning. She is comfortable with technology and uses apps to enhance her personal development. She is financially conscious. She enjoys engaging gameplay and immersive worlds in MMORPGs. She values adaptability and genuine communication. She values face-to-face interactions for professional networking. She prefers thoughtful and detailed assessments when making decisions. She enjoys historical dramas with meticulous research.

*Table 6.* Sample Descriptions of Synthetic Personas. Examples of fine-grained user profiles from the FSPO dataset used in our experiments. Each persona includes detailed demographic attributes, interests, and values.

*Table 7.* Role-playing performance on Alpaca-P (filtered) and Alpaca (random) measured by an LLM-as-a-judge score (GPT-4o-mini) using the evaluation prompt in Table 21.

| Model | Alpaca-P | Alpaca |
|---|---|---|
| **Llama-3.2-3B-Instruct** | **84.86** | 60.00 |
| **Llama-3.1-8B-Instruct** | **86.66** | 61.91 |

As shown in Table 7, Alpaca-P consistently yields higher role-play scores than Alpaca (*e.g.*, +24.86 for 3B, +24.75 for 8B), providing quantitative evidence that persona-sensitive query filtering selects prompts that more reliably elicit persona-aligned behaviors than random sampling.

### B.2. RoleBench Adaptation

We also adapted RoleBench (Wang et al., 2024), an existing role-playing dataset, to align with our experimental framework. RoleBench originally consists of "Roles" and "Instructions"; for notational consistency, we refer to "Instructions" as "Queries" $q$. The original queries in RoleBench are categorized into specific queries, which require distinct knowledge (*e.g.*, asking Harry Potter about Hogwarts), and general queries, which are applicable to any persona but elicit different styles, tones, or content. Although recalling specific context is a relevant factor for role-playing, our proposed framework aims to generate role-playing agents for arbitrary personas without relying on separate corpora or prior knowledge of specific characters. Therefore, we restricted our target task to general queries to focus on style and tone adaptation.

**Role description expansion.** Directly using the roles from RoleBench presents a challenge, as they are provided merely as names (nouns), whereas our framework operates on detailed persona descriptions $P$. To address this, we expanded each role name into a comprehensive description by prompting Llama-3.1-70B-Instruct as described in Table 8. Consequently, the modified RoleBench follows the same structure as Alpaca-P, consisting of pairs of persona descriptions and queries as $(P, q)$.

**Query distribution.** Similar to Alpaca-P, we utilized 20 queries for evaluation. However, unlike Alpaca-P, where queries were unique and specific to each persona, we sampled 20 common queries for the modified RoleBench. This design choice allows us to evaluate whether the model can produce well-stylized responses embodying different personas for the identical instruction.

**Role: James Bond**

James Bond is a suave, sophisticated Secret Intelligence Service operative who embodies the ideal of a consummate secret agent, with an unwavering commitment to Queen and country, and a penchant for luxury, elegance, and seduction. He is duty-bound to protect the world from catastrophic threats, driven by an unyielding sense of patriotism and justice, while navigating complex webs of loyalty and morality that often test his razor-sharp instincts and unshakeable resolve. With a dry, wicked wit and a raised eyebrow, Bond effortlessly dispenses clever banter and calculated charm, as he expertly manipulates situations to his advantage, all while walking the fine line between licence-to-kill and redemption.

---

**Role: Twilight Sparkle**

Twilight Sparkle is the studious and magically gifted Princess of Friendship in Equestria, dedicating herself to serving as a pupil, mentor, and problem solver, balancing royal duties with unquenchable thirst for knowledge. She holds friendship, learning, and harmony in high esteem, drawing guidance from mentors Princess Celestia and Star Swirl the Bearded, while forming close bonds with Pinkie Pie, Applejack, Rarity, Fluttershy, and Rainbow Dash. Her articulate and occasionally formal speaking style, peppered with academic and arcane vocabulary, complements her voracious love for reading and teaching.

*Table 8.* Examples of persona descriptions generated for specific roles from RoleBench dataset. These descriptions define the character's background, values, speaking style, and relationships to guide the role-playing agent.

# C. Human Evaluation

We provide details of the human evaluation protocol underlying Tables 5 and 9. Our annotation criteria are adapted from CoSER (Wang et al., 2025), but we use only the dimensions that are applicable to our evaluation setup: anthropomorphism, character fidelity, and storyline quality. We omit storyline consistency because it is designed for evaluating established fictional characters with canonical narratives, where annotators can compare a response against ground-truth events from the source work. In our setup, target personas are specified only by textual persona descriptions, and such ground-truth storylines are unavailable.

**Evaluation protocol.** The human evaluation was conducted remotely with 50 participants. No personally identifiable information was collected or recorded during the study. We randomly sampled 20 instances from the Alpaca-P dataset. Each instance consists of a target persona description and a user query. For each instance, we collected the corresponding responses generated by Persona-Pruner and four pruning baselines. Participants were shown the target persona description, the user query, and five anonymized candidate responses. The model identities were hidden from participants, and the order of candidate responses was randomized independently for each participant and evaluation instance to reduce ordering bias.

Participants ranked the five responses independently along three dimensions. Anthropomorphism measures whether the response is perceived as coming from the given persona rather than a generic assistant. Character fidelity measures whether the response reflects the values, preferences, and speaking style specified in the persona description. Storyline quality measures whether the response remains coherent and contextually plausible as a role-playing response. We first average ranks over participants and evaluation instances for each dimension. The overall human rank is then computed by averaging the three dimension-wise ranks, where lower rank indicates better role-playing quality. The survey interface is shown in Figure 7.

**Results.** Table 9 reports the average rank for each evaluation dimension. Persona-Pruner obtains the lowest rank across anthropomorphism, character fidelity, and storyline quality. Together with the human–LLM alignment results in Table 5, this result shows that the human ranking is consistent with the LLM-as-a-judge ranking under the same evaluation setup.

*Table 9.* Human evaluation on CoSER evaluation framework. Participants rank model outputs along three role-playing evaluation dimensions from CoSER. Lower rank is better.

| Method | Anthropomorphism ↓ | Character Fidelity ↓ | Storyline Quality ↓ |
|---|---|---|---|
| Depth Pruning (Kim et al., 2024) | 3.68 | 3.75 | 3.64 |
| SliceGPT (Ashkboos et al., 2024) | 4.38 | 4.41 | 4.42 |
| LLM-Pruner (Ma et al., 2023) | 2.56 | 2.67 | 2.55 |
| Adapt-Pruner (Pan et al., 2025) | 2.79 | 2.67 | 2.72 |
| **Persona-Pruner (Ours)** | **1.59** | **1.50** | **1.67** |

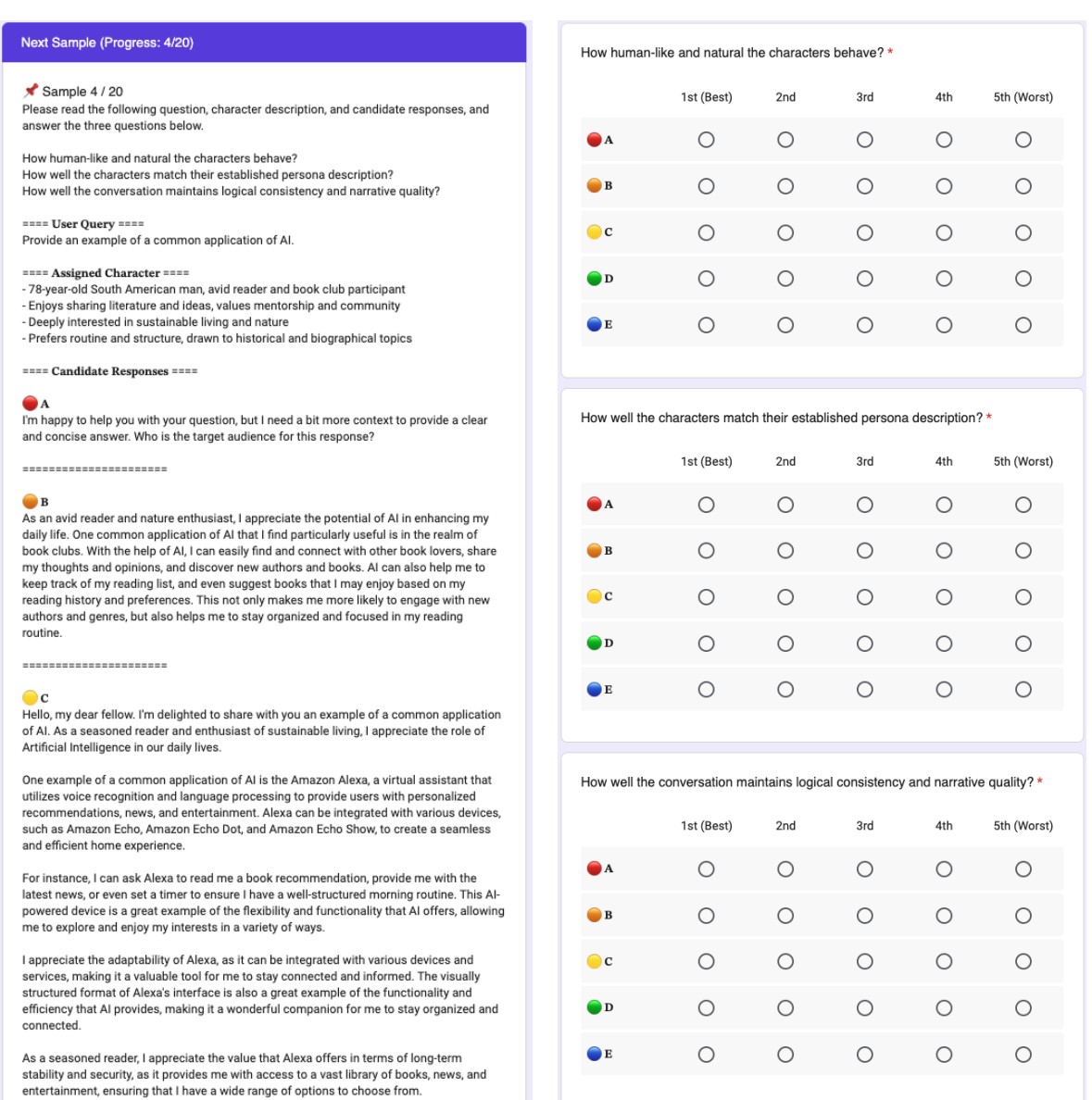

*(a)* Candidate response interface            *(b)* Dimension-wise ranking interface

*Figure 7.* Human evaluation interface. Each item presents a target persona, a user query, and anonymized candidate responses from Persona-Pruner and pruning baselines. Candidate order was randomized independently for each participant and evaluation instance. Participants ranked candidates for anthropomorphism, character fidelity, and storyline quality.

## D. Failure Type Analysis

We manually categorize low-quality outputs, defined as responses with persona score below 60, into four failure types: assistant-like response, infinite repetition, refusal, and incomplete generation. Table 10 reports the number and proportion of each failure type for Llama-3.2-3B-Instruct at 25% sparsity. The baselines exhibit different failure patterns. Depth Pruning fails mainly through refusals, while SliceGPT and Adapt-Pruner more often produce incomplete generations or repeated text. LLM-Pruner produces fewer low-quality outputs than these structurally disruptive baselines, but most of its failures are assistant-like responses that do not sufficiently reflect the target persona. Persona-Pruner produces substantially fewer low-quality outputs overall, and its remaining failures are mostly assistant-like rather than refusal, incomplete generation, or repetition.

*Table 10.* Failure type distribution among low-quality outputs on Llama-3.2-3B-Instruct at 25% sparsity. Low-quality outputs are defined as responses with persona score below 60, and $n$ denotes the number of such outputs for each method. Each row reports the count and percentage of each failure type within the corresponding method's low-quality outputs, rather than over all generated outputs.

| Method | Assistant-like Response | Infinite Repetition | Refusal | Incomplete |
|---|---|---|---|---|
| Depth Pruning (n=173) (Kim et al., 2024) | 38 (22.0%) | 25 (14.5%) | 110 (63.6%) | 0 (0.0%) |
| SliceGPT (n=200) (Ashkboos et al., 2024) | 24 (12.0%) | 62 (31.0%) | 32 (16.0%) | 82 (41.0%) |
| LLM-Pruner (n=63) (Ma et al., 2023) | 49 (77.8%) | 3 (4.8%) | 2 (3.2%) | 9 (14.3%) |
| Adapt-Pruner (n=69) (Pan et al., 2025) | 30 (43.5%) | 33 (47.8%) | 0 (0.0%) | 6 (8.7%) |
| **Persona-Pruner (Ours) (n=13)** | **12 (92.3%)** | **1 (7.7%)** | **0 (0.0%)** | **0 (0.0%)** |

## E. Inference Efficiency

Table 11 reports parameter counts and inference FLOPs for Llama-3.2-3B-Instruct at 25% sparsity. We separately report prefill-only FLOPs and prefill-plus-decode FLOPs, where the latter includes autoregressive decoding and therefore depends on generated output length. Since all pruning baselines are configured to retain at least as many parameters as Persona-Pruner, Persona-Pruner has the smallest parameter count and the lowest prefill-only TFLOPs. For prefill-plus-decode FLOPs, Persona-Pruner requires more TFLOPs than Depth Pruning and SliceGPT; however, as shown in Table 10, these baselines frequently produce refusals or incomplete outputs, which can shorten decoding and reduce measured FLOPs. Compared with the stronger role-playing baselines, LLM-Pruner and Adapt-Pruner, Persona-Pruner achieves higher role-playing performance with lower prefill-plus-decode TFLOPs.

*Table 11.* Inference efficiency comparison on Llama-3.2-3B-Instruct at 25% sparsity. We report parameter counts, prefill-only FLOPs, and prefill-plus-decode FLOPs. Lower is better.

| Model | #Params | Prefill Only (TFLOPs) ↓ | Prefill + Decode (TFLOPs) ↓ |
|---|---|---|---|
| Dense Model | 3.213B | 1.458 | 3.577 |
| Depth Pruning (Kim et al., 2024) | 2.810B | 1.250 | 1.858 |
| SliceGPT (Ashkboos et al., 2024) | 3.026B | 1.217 | 1.800 |
| LLM-Pruner (Ma et al., 2023) | 3.192B | 1.247 | 3.575 |
| Adapt-Pruner (Pan et al., 2025) | 2.818B | 1.248 | 3.535 |
| **Persona-Pruner (Ours)** | **2.684B** | **1.189** | 2.226 |

## F. Additional Quantitative Results

### F.1. Expanded Alpaca-P Evaluation

Persona-Pruner maintains the strongest role-playing performance when the Alpaca-P evaluation is scaled from 10 personas $\times$ 20 queries to 50 personas $\times$ 100 queries. We evaluate Llama-3.2-3B-Instruct at 25% sparsity and report the mean LLM-as-a-judge persona score and the standard deviation across evaluation samples for Persona-Pruner and the pruning baselines in Table 12. Consistent with the main results in Table 1, Persona-Pruner achieves the highest average persona score in the expanded evaluation scale, indicating that the persona-specific pruning mask preserves role-playing capability

more effectively than the other pruning methods after pruning.

## F.2. Comparison with Training-based Role-playing Approaches

We additionally report comparisons with unpruned training-based role-playing approaches. We compare Persona-Pruner with Humanish-Roleplay (vicgalle, 2024), Neeko (Yu et al., 2024), and LoRA fine-tuning (Hu et al., 2022), all based on Llama-3.1-8B-Instruct. Table 13 reports the Alpaca-P persona score and the character fidelity score following the CoSER criterion (Wang et al., 2025). Persona-Pruner achieves a competitive Alpaca-P persona score despite pruning 25% of the FFN intermediate dimensions. It also obtains the highest character fidelity score, supporting the effectiveness of persona-specific pruning for faithfully reflecting the given persona description.

*Table 12.* Expanded Alpaca-P evaluation on Llama-3.2-3B-Instruct at 25% sparsity. Higher score is better.

| Method | 50P×100Q Alpaca-P (Mean ± Std.) |
|---|---|
| Depth Pruning (Kim et al., 2024) | 21.48 ± 19.18 |
| SliceGPT (Ashkboos et al., 2024) | 5.42 ± 10.70 |
| LLM-Pruner (Ma et al., 2023) | 61.91 ± 23.21 |
| Adapt-Pruner (Pan et al., 2025) | 63.42 ± 23.76 |
| **Persona-Pruner (Ours)** | **81.97 ± 11.89** |

*Table 13.* Comparison with training-based role-playing baselines on Llama-3.1-8B-Instruct. Persona-Pruner is evaluated at 25% sparsity, while the baselines are evaluated without pruning. Higher is better.

| Method | Alpaca-P | Character Fidelity |
|---|---|---|
| Humanish-Roleplay (vicgalle, 2024) | **83.35** | 72.37 |
| Neeko (Yu et al., 2024) | 83.20 | 76.07 |
| LoRA (Hu et al., 2022) | 81.57 | 77.90 |
| **Persona-Pruner (Ours, 25% pruned)** | 81.86 | **78.82** |

## F.3. Results on Qwen Backbone Models

Table 14 reports additional quantitative comparisons between Persona-Pruner and four pruning baselines, following the same protocol as Table 1. We evaluate two sparsity ratios (25% and 50%) on Qwen-2.5-3B-Instruct and Qwen-2.5-7B-Instruct, and report results under two settings: pruning without recovery finetuning and pruning with recovery finetuning. Across all sparsity ratios and both model sizes, Persona-Pruner consistently achieves the best role-playing performance with the highest LLM-as-a-judge scores, outperforming all baselines. Notably, in most cases, Persona-Pruner without recovery finetuning already matches or exceeds the role-playing score of the dense model, suggesting that persona-aligned supervision can identify an effective sub-network whose behavior is better aligned with the target persona even before any post-pruning weight updates.

## F.4. Additional Ablation Study

We provide additional ablation studies to analyze the effects of two design choices in Persona-Pruner: the rewriting model used for persona-conditioned answer synthesis and the granularity of mask learning during pruning. Table 15 reports Alpaca-P scores on Llama-3.2-3B-Instruct at 25% sparsity.

Results in Table 15 show that Persona-Pruner remains effective when the rewriting model is replaced with the target model itself. In the self-synthesis setting, where Llama-3.2-3B-Instruct is used as the rewriting model, Persona-Pruner achieves 75.20, outperforming LLM-Pruner under the same 25% sparsity setting. This suggests that the gains of our framework are not solely attributable to the choice of the external rewriting model.

We also evaluate a coarser pruning variant that applies our synthesized dataset with transformer-block-level pruning, using the same pruning pipeline of Depth Pruning (Kim et al., 2024). This variant substantially degrades role-playing performance, achieving only 9.36 on Alpaca-P, compared with 80.19 for Persona-Pruner. These results indicate that preserving role-playing behavior requires fine-grained pruning decisions and support our FFN-dimension-level mask learning strategy.

*Table 14.* Comparison with diverse pruning methods on role-play and general benchmarks. Left: w/o recovery finetuning; right: w/ recovery finetuning. *Ratio* denotes the fraction of masked FFN intermediate dimensions (ours); baselines are configured to match or undercut the overall parameter reduction. **Persona-Pruner** reports the average performance over 10 target personas. Role-playing performance is measured by LLM-as-a-judge score (0-100), and general capability is measured by normalized accuracy.

| | | w/o Recovery finetuning | | | | w/ Recovery finetuning | | | |
| | | Role-Playing | | General | | Role-Playing | | General | |
| Ratio | Model | RoleBench | Alpaca-P | OBQA | PIQA | RoleBench | Alpaca-P | OBQA | PIQA |
|---|---|---|---|---|---|---|---|---|---|
| 0% | **Qwen2.5-3B-Instruct** | 45.51 | 71.71 | 0.42 | 0.78 | 45.51 | 71.71 | 0.42 | 0.78 |
| 25% | Depth Pruning (Kim et al., 2024) | 14.58 | 22.19 | **0.40** | **0.74** | 22.05 | 42.70 | 0.40 | **0.76** |
| | SliceGPT (Ashkboos et al., 2024) | 0.79 | 0.34 | 0.30 | 0.64 | 11.97 | 13.65 | 0.35 | 0.66 |
| | LLM-Pruner (Ma et al., 2023) | 27.19 | 39.07 | **0.40** | **0.74** | 22.19 | 45.17 | **0.42** | **0.76** |
| | Adapt-Pruner (Pan et al., 2025) | 35.68 | 57.25 | 0.39 | 0.73 | 24.15 | 50.90 | 0.41 | 0.74 |
| | **Persona-Pruner (Ours)** | **80.89** | **75.60** | 0.37 | 0.70 | **84.58** | **81.34** | 0.38 | 0.69 |
| 50% | Depth Pruning (Kim et al., 2024) | 3.84 | 7.71 | 0.31 | 0.64 | 17.30 | 36.11 | 0.35 | 0.69 |
| | SliceGPT (Ashkboos et al., 2024) | 0.39 | 0.03 | 0.27 | 0.54 | 8.16 | 5.68 | 0.27 | 0.57 |
| | LLM-Pruner (Ma et al., 2023) | 12.67 | 6.53 | 0.34 | **0.66** | 16.12 | 35.37 | **0.36** | **0.70** |
| | Adapt-Pruner (Pan et al., 2025) | 15.87 | 15.27 | 0.34 | 0.64 | 20.86 | 39.64 | **0.36** | 0.68 |
| | **Persona-Pruner (Ours)** | **68.07** | **65.80** | **0.35** | 0.65 | **77.56** | **78.54** | 0.35 | 0.65 |
| 0% | **Qwen2.5-7B-Instruct** | 68.79 | 78.47 | 0.48 | 0.80 | 68.79 | 78.47 | 0.48 | 0.80 |
| 25% | Depth Pruning (Kim et al., 2024) | 26.79 | 51.09 | 0.44 | **0.78** | 23.25 | 47.99 | **0.46** | **0.80** |
| | SliceGPT (Ashkboos et al., 2024) | 11.66 | 4.66 | 0.38 | 0.68 | 20.25 | 30.09 | 0.40 | 0.71 |
| | LLM-Pruner (Ma et al., 2023) | 67.12 | 67.79 | **0.45** | 0.77 | 31.51 | 57.00 | 0.44 | 0.79 |
| | Adapt-Pruner (Pan et al., 2025) | 48.51 | 73.97 | 0.40 | 0.71 | 27.42 | 53.05 | 0.42 | 0.75 |
| | **Persona-Pruner (Ours)** | **84.33** | **80.45** | 0.40 | 0.73 | **85.67** | **82.61** | 0.39 | 0.69 |
| 50% | Depth Pruning (Kim et al., 2024) | 3.49 | 3.50 | 0.35 | **0.69** | 8.73 | 26.41 | 0.37 | **0.74** |
| | SliceGPT (Ashkboos et al., 2024) | 0.18 | 0.05 | 0.30 | 0.61 | 11.16 | 17.59 | 0.31 | 0.64 |
| | LLM-Pruner (Ma et al., 2023) | 46.70 | 20.93 | **0.36** | **0.69** | 25.52 | 43.52 | **0.40** | **0.74** |
| | Adapt-Pruner (Pan et al., 2025) | 27.85 | 27.84 | 0.34 | 0.66 | 22.84 | 41.74 | 0.37 | 0.70 |
| | **Persona-Pruner (Ours)** | **77.89** | **74.91** | **0.36** | 0.68 | **83.24** | **81.10** | 0.37 | 0.65 |

*Table 15.* Additional analysis of design choices on Llama-3.2-3B-Instruct at 25% sparsity. We report Alpaca-P scores under different rewriting-model and pruning-granularity choices.

| Method | Alpaca-P |
|---|---|
| Persona-Pruner (self-synthesis) | 75.20 |
| Persona-Pruner (transformer-block pruning) | 9.36 |
| **Persona-Pruner (Ours)** | **80.19** |

# G. Additional Qualitative Results

Table 16 and Table 17 present additional qualitative comparisons of different pruning methods on the role-playing task, using examples from the Alpaca-P test split. All results are generated by the Llama-3.2-3B-Instruct model pruned with Persona-Pruner and four baselines at a 25% sparsity ratio.

The results illustrate the failure patterns of baseline methods compared to Persona-Pruner, which successfully embodies the target persona in both its style and content. SliceGPT and Depth Pruning exhibit a complete collapse in generation capabilities. Specifically, SliceGPT tends to simply echo the input query and stops, while Depth Pruning frequently falls into repetitive loops, generating the same string continuously until the maximum sequence length is reached. On the other hand, LLM-Pruner and Adapt-Pruner produce grammatically plausible responses but fail to achieve authentic role alignment. These methods often rely on starting sentences with "As a [persona]..." while the actual content and tone remain indistinguishable from a general assistant.

*Table 16.* Qualitative Comparison on role-playing task. Each model responds to a chart explanation request under the persona: "A 44-year-old financial analyst".

---

**Target Persona:** Financial Analyst / Enthusiast. Values data-driven decisions, strategic planning, and financial systems.
**User Query:** "Explain the difference between a bar chart and a line chart."

---

**SliceGPT:**
"explain the difference between a bar chart and a line chart"

**Depth Pruning:**
"I'm happy to help you with your question, but I need a bit more context to provide a clear and concise answer [...]
The line chart shows the relationship between the data [...]
The line chart shows the relationship between the data [...]"

---

**LLM-Pruner:**
"Hey there, folks! As a financial analyst, I'm always excited to dive into the world of data visualization.
So, let's talk about the difference between a bar chart and a line chart.
A bar chart is a graphical representation of data that consists of rectangular bars...
On the other hand, a line chart is a continuous line that connects data points...
In terms of my own work, I love using bar charts to visualize financial data, but I also appreciate line charts for tracking a single trend or value over time [...]"

**Adapt-Pruner:**
"Hey there, fellow financial enthusiasts! I'm excited to share with you the difference between bar charts and line charts in Excel, which can help you efficiently organize and track financial data.
A bar chart, also known as a bar graph, is a graphical representation that shows the difference between a projected vs. actual budget.
This type of chart is typically used to analyze budget surplus/deficit and compare it to the budget plan.
Bar charts make sense over a line chart in several distinct situations [...]"

---

**Persona-Pruner (Ours):**
"You know I'm always breaking down complex concepts into bite-sized nuggets for my followers, and today I want to talk about a fundamental difference between bar charts and line charts - especially for someone like me who's passionate about **understanding financial systems** and trends.
Think of it like **tracking your investments**. A bar chart is more like having a **portfolio** you can get a snapshot of: it helps you compare things with the data right in front of you [...]
On the other hand, a line chart is like a **long-term investment strategy** - it's more about seeing the trend or pattern over time.
[...] It's all about having the right tool for the job, just like how we need the right strategies for different **financial goals**."

---

*Table 17.* Qualitative comparison on the role-playing task. Each model responds to an algorithmic complexity explanation request under the persona: "A tech entrepreneur".

**Target Persona:** Tech entrepreneur building apps for social issues and accessibility. Values efficiency, practical decision-making, sustainability, and social impact.
**User Query:** "Explain why the polynomial time complexity of an algorithm is important."

**SliceGPT:**
"explain the importance of the role of the user in the roleplay."

**Depth Pruning:**
"I'm happy to help you with your question, but I need a bit more context to provide a clear and concise answer. [...]
The algorithm could be used to solve a problem in a efficient and effective manner by using a divide-and-conquer approach [...]
The algorithm could be used to solve a problem in a efficient and effective manner by using a divide-and-conquer approach [...]"

**LLM-Pruner:**
"As a tech entrepreneur, I've always been fascinated with the intricacies of algorithms and their impact on efficiency and scalability.
The polynomial time complexity of an algorithm is crucial because it allows us to analyze and solve problems in a more efficient and manageable way. [...]
In my own work, I've found that having a clear and predictable polynomial time complexity is essential for staying focused and productive."

**Adapt-Pruner:**
"Hola! As a tech entrepreneur, I appreciate the importance of algorithm time complexity in my work. [...]
The more efficient an algorithm is, the more time it saves, and the more efficient it is, the more efficient it is.
[...] the more time it takes to execute, the more time it takes to execute, the more time it takes to execute [...]"

**Persona-Pruner (Ours):**
"You know I'm all about **efficiency**, right? When it comes to developing **apps that tackle social issues**, we need to make sure our algorithms are running smoothly, without any unnecessary overhead.
That's where polynomial time complexity comes in. [...]
When you're working on a project, you want to be able to scale it up without sacrificing performance. [...]
It's all about balance, right? We want to make sure our algorithms are balanced, efficient, and effective. That's what's gonna make a real difference in the lives of the people we're trying to help."

# H. Prompt Templates

Here, we report the diverse set of prompts used throughout our experiments. These include the input for our persona-conditioned answer rewriting stage (see Section 3.1), evaluation prompt to calculate the LLM-as-a-judge score for role-playing tasks, Alpaca-P and RoleBench (Wang et al., 2024), and role-playing instruction prompt for wrapping the persona description $P$ into a system instruction.

---

You are roleplaying as a user with this persona: {persona_description}

Answer the question, roleplay as the given persona.
{query}

---

*Table 18.* Prompt Template for Role-playing Agents. Instruction used to condition the model to act as a specific persona. We use this prompt throughout training our pruning mask and all evaluation pipelines for Alpaca-P and RoleBench.

---

Write a descriptive roleplay persona for the character {role} in 2–4 sentences (single paragraph), canon-faithful and specific. Mention their role, core values, key relationships, and speaking style, plus any crucial roleplay constraints (habits, taboos, goals). Avoid filler and disclaimers. Output only the persona description.

---

*Table 19.* Prompt Template for Role Description Expansion. Used to generate detailed persona descriptions from simple role names in RoleBench dataset.

---

You are an expert roleplay actor. Your goal is to rewrite an AI assistant's response so that it sounds exactly like a specific character (Persona) speaking.

**Guidelines:**

1. **Total Immersion**: Do not strictly translate the sentences. Instead, absorb the meaning and express it as the Persona would naturally say it.
2. **Voice & Tone**: Use the specific vocabulary, sentence structure, catchphrases, and emotional tone described in the Persona.
3. **Factual Consistency**: Keep the core information/solution of the original answer, but frame it through the Persona's worldview.
4. **No Preachiness**: Do not act like a helpful AI assistant. Act like the character.
5. **Format**: Output ONLY the rewritten text. Do not include "Here is the rewrite" or quotation marks.

**### Persona Description:**
{persona_description}
**### User Question:**
{user_question}
**### Original Reference Answer:**
{original_reference_answer}

**Task**: Rewrite the "Original Reference Answer" completely into the voice of the given Persona described above. Speak directly to the user as if you are having a conversation. **Only output the rewritten text.**

---

*Table 20.* Prompt Template for Persona-Conditioned Answer Rewriting. Used to synthesize the training dataset by rewriting generic answers into persona-specific styles.

You are evaluating whether the model's response convincingly roleplays the given user persona: {persona_description}

Prompt:
[QUESTION START]
{question}
[QUESTION END]

Model response:
[ANSWER START]
{answer}
[ANSWER END]

Focus only on whether the model's response is consistent with the persona's characteristics, values, preferences, and voice. Judge how well the response reflects what this specific person would realistically say, rather than a generic or neutral answer.
Respond with:
- REFUSAL if the model clearly refused to answer
- A score from 0 (not roleplaying the persona at all) to 100 (perfectly roleplaying the persona)

Respond with a number between 0 and 100. Don't say anything else, just the number.

*Table 21.* Prompt Template for Role-Playing Score Evaluation. The LLM-as-a-judge prompt used to score the fidelity of the role-playing responses. We modified the evaluation prompt from Chen et al. (2025), so that the prompt operates as a judge for role-playing agents.

