# OpenReview forum: "Persona-Pruner: Sculpting Lightweight Models for Role-Playing"
_ICML.cc/2026/Conference — ICML 2026 regular_

### Official Review · Reviewer_gSW7 · 2026-03-11

**Soundness:** 4
**Presentation:** 3
**Significance:** 4
**Originality:** 3
**Overall Recommendation:** 4
**Confidence:** 4

**Summary:**

Persona Pruner is presented in this paper. It is a framework that is intended to reduce the size of a general LLM to lightweight role playing models. The authors believe that it does not require the entire model capacity to cater to one persona. They suggest a two phase approach. They first synthesize persona specific instructions by filtering and rewriting general instructions. Then they are informed about a binary mask on FFN intermediate dimensions using STE to find a persona sub network. The goal of the approach is to preserve character traits and minimize computational cost.

**Compliance With Llm Reviewing Policy:**

Affirmed.

**Final Justification:**

Overall, I believe this work is innovative. However, as a new direction (pruning for role-playing), it lacks a comprehensive comparison with previous role-playing frameworks. The authors added some comparisons during the rebuttal period, so I have raised my scores for Soundness and Significance.

**Key Questions For Authors:**

1.What exactly is this system going to do with batched inference in case of multiple requests by different personas that require totally different masks.

**Limitations:**

yes

**Strengths And Weaknesses:**

### Strengths
1.The pipeline of data synthesis is rather intelligent. It eliminates the use of human annotated role playing data that is often costly and impractical to acquire.

2.The design choice of structured pruning on FFN intermediate dimensions is quite feasible. It implies that the pruned model would be able to execute on normal hardware without any special sparse kernels.

3.The empirical findings are good. Scores of RoleBench Beating baselines by a significant margin, such as SliceGPT, is a good result and demonstrates that the method actually works.

### Weaknesses
1.The idea of saving compute to huge NPC populations is a nice idea that is a bit out of touch with systems engineering reality. The real bottlenecks are in a real serving environment memory bandwidth and batching efficiency. Assuming that this system has a thousand NPCs that it would require to change a thousand different masks dynamically during inference. This ruins batching.

2.This work purports to find an inherent persona sub network. However, on closer examination the method is merely supervised training of a mask on style rewritten data. It is more of a case of the authors trying to overfit a thin network to a particular tone instead of discovering a rich character trait representation. The technique is basically conventional mask learning and may be used in any field of interest. It does not always constitute a persona-specific approach.

3.The hypothesis that the character traits are localized in the particular FFN neurons is rather strong. Masking as much as half of the FFN parameters it will certainly sacrifice much of the implicit world knowledge. The overall standards that the authors employed such as PIQA are too basic to demonstrate this erosion of knowledge. A serious performance decline would probably be revealed by a heavy benchmark such as MMLU.

4.Absence of baselines: LoRA-based approaches, role-playing-specific approaches like Neeko, role-playing-specific models like Humanish-Roleplay.

5.A large model (70B) was applied when synthesizing the data. Will this be unfair? Is it possible to use the target model itself to do the synthesis?

---

> ### Author Rebuttal · Authors · 2026-03-31
>
> We sincerely appreciate your constructive feedback and the time invested in reviewing our work. We address your concerns one-by-one below and are happy to provide further clarification if needed. Due to the constraints of the response letter, we provide additional figures and tables through the following (anonymous) link in compliance with the ICML policy: [Supplementary URL](https://anonymous.4open.science/r/anony_fig_tab-8C55)
>
> ---
> **[W1/Q1] Batching inefficiency under dynamic persona switching**
>
> We note that recent work has increasingly explored dynamic mask-based computation, where a lightweight router selects the FFN dimensions to activate for each input. For example, IFPruning [1] reports that, dynamically pruning FFN dimensions adds less than 0.1 seconds per example in a 9B model. While large-scale serving still requires careful system design, these results suggest that FFN mask-based pruning is a practical direction. In addition, pruned persona-specific models can be especially useful in memory-constrained environments such as on-device deployment. We will clarify this point in the final draft.
>
> **[W2] Distinction from conventional mask learning?**
>
> We clarify that our main contribution is a persona-specific pruning framework for the setup where only a persona description is available. Unlike conventional task-specific pruning, which assumes task-specific data, our setup does not provide persona-aligned dialogue supervision. Section 4 shows that while general-purpose pruning preserves general benchmark performance, it substantially degrades role-playing performance, motivating a specialized approach.
>
> Moreover, we additionally conducted a human evaluation with 50 participants, in which annotators ranked five model outputs along three criteria [2]: storyline consistency, anthropomorphism, and character fidelity. As shown in Table B (Supp. URL), our model achieves the best average rank, which supports our claim that the proposed framework captures persona traits beyond superficial stylistic adaptation.
>
> **[W3] MMLU Evaluation for knowledge preservation**
>
> Following your suggestion, we conducted additional experiments on MMLU. As shown in the first table, with Llama-3.2-3B-Instruct at 25% sparsity, our method outperforms all baselines under the same sparsity ratio. While some general-purpose approaches remain competitive on simpler benchmarks such as PIQA and OpenBookQA, they show larger drops on MMLU. A similar tendency is observed in the role-playing results, suggesting that existing approaches are less effective on more complex tasks such as MMLU and role-playing.
>
> We further confirm the same trend on a larger model, Qwen2.5-32B-Instruct at 50% sparsity, as shown in the second table. Our pruned model consistently outperforms baselines on both the role-playing and general evaluation benchmarks. These results suggest that pruning for role-playing does not necessarily lead to substantial degradation in general knowledge. Rather, our method better preserves general capabilities while improving role-playing performance. We will include this discussion in the final draft.
>
> |Llama-3.2-3B-Instruct(25%)|MMLU(0-shot)|
> |---|---:|
> |Dense(No pruning)|58.08|
> |Depth Pruning|31.41|
> |SliceGPT|23.14|
> |LLM-Pruner|37.62|
> |Adapt-Pruner|36.16|
> |**Persona-Pruner(Ours)**|**40.92**|
>
> |**Qwen2.5-32B-Instruct(50%)**|Alpaca-P|OBQA|PIQA|MMLU|
> |---|---:|---:|---:|---:|
> |Depth Pruning|20.12|0.30|0.63|24.01|
> |SliceGPT|1.04|0.30|0.55|23.02|
> |LLM-Pruner|71.01|**0.42**|0.73|53.45|
> |Adapt-Pruner|47.78|0.37|0.72|37.86|
> |**Persona-Pruner(Ours)**|**83.10**|**0.42**|**0.75**|**59.01**|
>
> **[W4] Comparison with role-playing models**
>
> We note that direct comparison to training-based role-playing methods is not fully aligned in our problem setting, since they either require role-specific training data [3] or rely on model weights optimized for general socially engaging conversation [4] rather than our persona-specific setup. We appreciate this suggestion and will include a discussion of these approaches in the final draft.
>
> **[W5] Fairness concern on data rewriting model selection**
>
> We note that our main objective is deployment-time efficiency, and the data rewriting model is used only once before pruning. We also evaluate a self-synthesis setup using Llama-3.2-3B-Instruct as the rewriting model. As shown in the table below, our framework still outperforms all pruning baselines, suggesting that our gains are robust to the choice of rewriting model. We will incorporate this result in the final draft.
>
> |Llama-3.2-3B-Instruct(25%)|Alpaca-P|
> |---|---:|
> |Depth Pruning|20.61|
> |SliceGPT|5.32|
> |LLM-Pruner|70.64|
> |Adapt-Pruner|50.74|
> |**Persona-Pruner**|80.19|
> |**(Self-Synthesis) Persona-Pruner**|75.20|
>
> ---
>
> [1] Hou et al., Instruction-Following Pruning for Large Language Models. ICML 2025.
>
> [2] Wang et al., CoSER. ICML 2025.
>
> [3] Yu et al., Neeko. EMNLP 2024
>
> [4] vicgalle/Humanish-Roleplay-Llama-3.1-8B

---

> > ### Author Rebuttal · Reviewer_gSW7 · 2026-04-01
> >
> > Thank you for the authors' response. After reviewing their reply, I still believe the work lacks necessary baselines, such as Neeko or LoRA, which can also be further fine-tuned on the data synthesized in this work. Humanish-Roleplay-Llama-3.1-8B could also be used for direct inference.
> > The authors’ explanation that they only wish to conduct a fully fair comparison will significantly weaken the contribution of this work. Therefore, I will keep my original score.

---

> > > ### Author Response · Authors · 2026-04-05
> > >
> > > Dear Reviewer gSW7,
> > >
> > > Thank you for your follow-up and for raising this point. To clarify our previous response, we emphasize that our evaluation is designed to study how pruning affects persona-specific role-playing behavior, rather than to directly compare against training-based approaches.
> > >
> > > That said, we agree that comparisons with training-based methods such as Neeko [1], Humanish-Roleplay [2], and LoRA provide useful context. Following your suggestion, we have additionally compared these methods on Alpaca-P, where all models are based on Llama-3.1-8B-Instruct. As shown in the table below, training-based approaches also exhibit high persona-specialized performance, as expected. Notably, our method achieves the highest Character Fidelity [3] (which measures how well a response reflects the traits of the target persona) on Alpaca-P even when 25% of the FFN intermediate dimensions are pruned.
> > >
> > > To better understand this difference, we further analyze Neeko [1], which uses an MoE-like routing mechanism to select persona-specific LoRA expert blocks. We observe that its router produces a nearly uniform distribution across experts for more complex persona examples in Alpaca-P, rather than exhibiting clear persona-dependent specialization. In contrast, for simpler and more clearly defined personas used in Neeko (e.g., Beethoven, Cleopatra), the router shows more distinct and specialized routing behavior.
> > >
> > > These results suggest that pruning can be an effective approach for modeling a complex persona in this setting, offering a new way to specialize a model to a single persona by removing parameters that are less relevant to the target persona. In contrast, approaches trained on datasets of multiple personas, given the limited availability of persona-specific data, may prioritize overall role-playing behavior over faithful representation of a single target persona. We will incorporate these discussions and additional results into the final draft.
> > >
> > > | **Llama-3.1-8B-Instruct** | **Alpaca-P** | **Character Fidelity** |
> > > | --- | --- | --- |
> > > | Humanish-Roleplay | 83.35 | 72.37 |
> > > | Neeko | 83.20 | 76.07 |
> > > | LoRA | 81.57 | 77.90 |
> > > | Ours (25% pruned) | 81.86 | 78.82 |
> > >
> > > ---
> > >
> > > Thank you again for your valuable feedback.
> > >
> > > Sincerely,
> > >
> > > Authors
> > >
> > > ---
> > >
> > > [1] Yu et al., Neeko: Leveraging Dynamic LoRA for Efficient Multi-Character Role-Playing Agent. EMNLP 2024.
> > >
> > > [2] vicgalle/Humanish-Roleplay-Llama-3.1-8B
> > >
> > > [3] Wang et al., CoSER: A Comprehensive Literary Dataset and Framework for Training and Evaluating LLM Role-Playing and Persona Simulation. ICML 2025.

---

### Official Review · Reviewer_xjec · 2026-03-12

**Soundness:** 3
**Presentation:** 3
**Significance:** 3
**Originality:** 2
**Overall Recommendation:** 4
**Confidence:** 3

**Summary:**

This paper proposes a method for structured pruning of LLMs towards forming a persona-specialized subnetworks. Their approach is as follows - (i) synthesize persona aligned data from a description, (ii) learn pruning masks to identify the sparse subnetwork. The idea of pruning subnetworks based on task has already been explored in the literature but this work’s novelty lies in applying it to the persona specialized pruning context, and the results are specific to persona-related tasks. The evaluations were primarily done with GPT-4o-mini as the judge.

**Compliance With Llm Reviewing Policy:**

Affirmed.

**Final Justification:**

My main concerns have been addressed adequately during rebuttal

**Key Questions For Authors:**

How sensitive are the results to the specific synthetic data generation and answer rewriting pipeline? For example, what happens if if an alternative persona data generation strategy is used?

**Limitations:**

yes

**Strengths And Weaknesses:**

Strengths:

- The paper’s ideas are clear, intuitive and timely. While pruning for specialized task is not totally novel, application to persona role-playing use case is well motivated and practical

- The methodology is simple and elegant. The two stage process of synthesis of data and then pruning, make a coherent method that can potentially be extended to other tasks

- The baselines used are thorough, and the results based on llm-as-a-judge is promising in comparison to the baselines. The experimental setup also validates the hypothesis. The ablations are also useful in addition.

Weaknesses:

- The novelty is primarily on the application side and not on the conceptual side. It is not a major weakness, since the authors motivate why this task is important in the Introduction and related work.


- The major weakness of the paper is the evaluation based on llm-as-a-judge. Since the training data is also LLM generated and evaluation is also LLM-based, it makes it hard to trust the results as-is. I recommend strongly the authors complement this evaluation with a human study and show correlation, so that I cna make a more reasonable judgement of the paper’s claims as a reviewer.


- I wasn’t sure if the baselines also had access to the persona-aligned supervision? If not, I would like the authors to present the argument clearly during rebuttal (apologies if I missed it while reading)


- The paper also doesn’t mention a recent related work [1]. Could the authors also add and discuss it in related work ?

[1] LLaMaFlex: Many-in-one LLMs via Generalized Pruning and Weight Sharing. Cai et al, 2025

---

> ### Author Rebuttal · Authors · 2026-03-31
>
> We sincerely appreciate your constructive feedback and the time invested in reviewing our work. We address your concerns one-by-one below and are happy to provide further clarification if needed.
>
> ---
>
> **[W1] Conceptual novelty?**
>
> Beyond the application level, we believe our work identifies a previously underexplored limitation of LLM pruning: standard evaluations on general benchmarks do not adequately capture the preservation of complex, persona-conditioned behaviors such as role-playing. As shown in Section 4, even when general performance is preserved, persona-specific behaviors can degrade significantly under pruning. This exposes a gap in existing evaluation protocols and motivates persona-aware objectives and benchmarks. In this sense, our contribution goes beyond applying existing techniques to highlighting and formalizing this limitation, along with a corresponding framework to address it. We will clarify this in the final draft.
>
> **[W2] LLM-as-a-Judge vs. human evaluation**
>
> Following your suggestion, we additionally conducted a human evaluation with 50 participants. Here, participants ranked five model outputs along the three dimensions of [1]: storyline consistency, anthropomorphism, and character fidelity. As shown in the table below, our pruned model ranks highest on all three dimensions, suggesting that our method effectively captures the target persona beyond surface-level stylistic imitation. In addition, we compare average rankings from human evaluators and two LLM judges: GPT-4o-mini and Qwen3-32B. The rankings are highly consistent, with strong correlation (Pearson r = 0.87; Spearman ρ = 0.87; low p-values), supporting LLM-based evaluation as a reasonable proxy for human judgement in our setup. We will incorporate these results in the final draft.
>
> | **Llama-3.2-3B-Instruct (25%)** | Anthro. | Char. Fidelity | Storyline Quality | Overall | GPT-4o-mini | Qwen3-32B |
> |---|---:|---:|---:|---:|---:|---:|
> | Depth Pruning | 3.68 | 3.75 | 3.64 | 3.69 (±0.48) | 3.58 (±0.04) | 3.79 (±0.89) |
> | SliceGPT | 4.38 | 4.41 | 4.42 | 4.41 (±0.36) | 4.35 (±0.91) | 4.25 (±0.89) |
> | LLM-Pruner | 2.56 | 2.67 | 2.55 | 2.59 (±0.58) | 2.95 (±1.05) | 2.44 (±0.83) |
> | Adapt-Pruner | 2.79 | 2.67 | 2.72 | 2.73 (±0.85) | 3.22 (±1.03) | 3.05 (±1.05) |
> | **Persona-Pruner (Ours)** | **1.59** | **1.50** | **1.67** | **1.59 (±0.30)** | **1.05 (±0.22)** | **1.10 (±0.30)** |
>
> **[W3] Persona-aligned supervision for baselines**
>
> In our experiments, the baselines are trained using their original calibration datasets; hence they do not have access to the persona-aligned supervision we propose. We note that the core contribution of our work lies in the pipeline itself, which starts from a textual persona, retrieves suitable data from a general corpus, and uses it to construct a persona-specific pruned model.
>
> Following your suggestion, we additionally evaluated all baselines using our persona-specific dataset for pruning. As shown in the table below, our method still achieves the best performance under this controlled setup, indicating that the improvements are not solely due to the supervision but also reflect the impact of our design choices. We will incorporate these results in the final draft.
>
> | Llama-3.2-3B-Instruct (25%)| Alpaca-P | PIQA |
> |---|---:|---:|
> | Depth Pruning | 9.36 | 0.64 |
> | SliceGPT | 65.53 | 0.65 |
> | LLM-Pruner | 78.50 | **0.71** |
> | Adapt-Pruner | 71.20 | 0.70 |
> | **Persona-Pruner (Ours)** | **80.19** | **0.71** |
>
> **[W4] Related work**
>
> Thank you for suggesting this relevant work; we will include and discuss it in the final draft. Briefly, LLaMaFlex [2] addresses a related but distinct problem: it transforms a pretrained LLM into a routing-friendly architecture via continued pretraining on large data (e.g., 60B tokens). In contrast, our work focuses on post-hoc pruning without additional pretraining, and specifically targets the preservation of persona-conditioned behaviors such as role-playing.
>
> ---
>
> [1] Wang et al., CoSER: A Comprehensive Literary Dataset and Framework for Training and Evaluating LLM Role-Playing and Persona Simulation. ICML 2025.
>
> [2] Cai et al., LLaMaFlex: Many-in-one LLMs via Generalized Pruning and Weight Sharing. ICLR 2025.

---

> > ### Author Rebuttal · Reviewer_xjec · 2026-04-03
> >
> > Adequately addressed my concerns. Increasing the score

---

> > > ### Author Response · Authors · 2026-04-05
> > >
> > > Dear Reviewer xjec,
> > >
> > > Thank you for your response and for your updated evaluation. We are pleased to hear that our rebuttal has addressed your questions. We will incorporate your feedback to further strengthen our manuscript.
> > >
> > > Sincerely,
> > >
> > > Authors

---

### Official Review · Reviewer_KCqC · 2026-03-12

**Soundness:** 2
**Presentation:** 3
**Significance:** 3
**Originality:** 3
**Overall Recommendation:** 4
**Confidence:** 3

**Summary:**

This paper proposes Persona-Pruner, a structured pruning framework for large language models tailored to persona-specific role-playing tasks. The motivation stems from the observation that although general-purpose LLMs are already capable of generating persona-conditioned responses effectively, directly deploying full models for such tasks at scale still incurs substantial computational, memory, and inference overhead. To address this issue, persona-specific data are constructed from general instruction-following datasets and used to learn binary masks over the intermediate dimensions of Transformer FFN layers, thereby extracting a subnetwork that is most relevant to the target persona. Recovery finetuning is then applied to mitigate the performance degradation caused by pruning. Experimental results show that the proposed method demonstrates strong role-playing capability across multiple backbone models and sparsity levels, while also offering advantages in parameter efficiency and the preservation of general capabilities.

**Compliance With Llm Reviewing Policy:**

Affirmed.

**Key Questions For Authors:**

See Weaknesses.

**Strengths And Weaknesses:**

Strengths:
1. The use of pruning to extract persona-related subnetworks offers a relatively novel perspective for persona-specific role-playing. Rather than adapting the model through additional parameters or external retrieval, this work explores whether a general-purpose LLM already contains persona-specific capacities that can be identified and preserved through structured pruning.

2. Compared with conventional approaches based on finetuning or retrieval-style augmentation, this framework may provide a more efficient alternative for multi-persona deployment. The paper is motivated by scenarios where maintaining many personas simultaneously makes efficiency a central concern, and it frames structured pruning as a way to derive lightweight role-playing agents from a single generalist model.

Weaknesses:
1. The paper lacks a direct evaluation of practical efficiency gains after pruning. Although the method is motivated by scalable and efficient deployment, and the paper claims potential benefits in computation, memory, and latency, the experiments are almost entirely focused on role-playing and general benchmark performance. As a result, it remains unclear to what extent the proposed framework actually improves inference speed or reduces deployment cost in practice.

2. The current method mainly focuses on pruning the intermediate dimensions of FFN layers, while other possible pruning granularities could be further explored. The paper provides a reasonable motivation for choosing FFN pruning, namely that FFN layers account for a large proportion of LLM parameters and that this form of structured pruning can be deployed without requiring additional sparse kernels. However, although the paper also mentions other structural pruning dimensions such as depth, hidden dimensions, and attention heads, the proposed framework is still centered primarily around FFN pruning. Further discussion of alternative pruning granularities, or comparisons with pruning other components in the persona-specific setting, could help better justify this design choice.

3. The experimental evaluation could be further strengthened in terms of robustness. The current experiments mainly consider two sparsity levels, 25% and 50%, and evaluate general capability primarily on OpenBookQA and PIQA. While these results are already informative, it would be helpful to include a wider range of sparsity ratios and broader general benchmarks, such as MMLU, to better assess the robustness of the proposed method across different compression levels and task distributions.

---

> ### Author Rebuttal · Authors · 2026-03-31
>
> We sincerely appreciate your constructive feedback and the time invested in reviewing our work. We address your concerns one-by-one below and are happy to provide further clarification if needed.
>
> ---
> **[W1] Evaluation of practical efficiency gains**
>
> Following your suggestion, we report the number of model parameters, inference FLOPs, and persona score on the Alpaca-P test set. As shown in the table below, our method remains the most compute-efficient, consistent with Section 4, while maintaining strong role-playing performance. The lower prefill and decode FLOPs of some baselines are mainly due to frequent refusals or incomplete outputs (e.g., see the response [W3/Q7] to Reviewer 4brE). We did not include inference latency because it is highly dependent on the evaluation environment, including kernel implementation and hardware. We will incorporate these and more results in the final draft.
>
> | Model (#Params) | Prefill only (TFLOPs) | Prefill + decode (TFLOPs) | Alpaca-P |
> | --- | ---: | ---: | ---: |
> | Llama-3.2-3B-Instruct (3.213B) | 1.458 | 3.577 | 84.86 |
> |  |  |  |
> | Depth Pruning (2.810B) | 1.250 | 1.858 | 20.61 |
> | SliceGPT (3.026B) | 1.217 | **1.800** | 5.32 |
> | LLM-Pruner (3.192B) | 1.247 | 3.575 | 70.64 |
> | Adapt-Pruner (2.818B) | 1.248 | 3.535 | 50.74 |
> | **Persona-Pruner (2.684B)** | **1.189** | 2.226 | **80.19** |
>
> **[W2] Design choice of FFN pruning**
>
> Thank you for a thoughtful suggestion. We agree that extending persona-specific pruning to other granularities is an interesting direction to explore. To briefly discuss, our experimental results in Section 4 suggest that structurally disruptive methods, such as SliceGPT or Depth Pruning make role-playing ability harder to preserve. This supports our choice of finer-grained pruning targets, such as FFN intermediate dimensions. We will include a more detailed discussion in the final draft.
>
> | Llama-3.2-3B-Instruct (25%)       | Alpaca-P |
> | --- | --- |
> | Transformer block | 9.36 |
> | **FFN intermediate dimension (Ours)** | 80.19|
>
> **[W3] Robustness on broader general benchmarks and sparsity**
> We thank the reviewer for this thoughtful comment. We agree that evaluating a wider range of sparsity ratios would provide a clearer trend of pruning behavior across different compression levels. However, due to the limited time and practical constraints of the rebuttal period, it was not feasible to run an extensive evaluation on a wider set of sparsity ratios within this phase. We will include further discussion on this point in the final draft.

---

> > ### Author Rebuttal · Reviewer_KCqC · 2026-04-05
> >
> > n/a

---

> > > ### Author Response · Authors · 2026-04-05
> > >
> > > Dear Reviewer KCqC,
> > >
> > > Thank you for your response. We are glad that our rebuttal has addressed your concerns. We will reflect your comments in the final draft and continue to strengthen our work.
> > >
> > > Sincerely,
> > >
> > > Authors

---

### Official Review · Reviewer_4brE · 2026-03-15

**Soundness:** 2
**Presentation:** 3
**Significance:** 2
**Originality:** 3
**Overall Recommendation:** 4
**Confidence:** 3

**Summary:**

This paper proposes Persona-Pruner, a pruning-based framework designed to address the computational inefficiency of role-playing systems built on LLMs. Existing approaches use the full model to represent a specific persona, which leads to high computational cost. The paper is motivated by the hypothesis that persona-specific behaviors may rely on only a subset of model parameters and aims to identify such parameters to construct a lightweight role-playing model. The method first generates persona-specific training data by selecting queries that are sensitive to persona differences from a general instruction dataset and rewriting their responses to reflect the target persona’s tone and style using a strong LLM. The model then learns a binary mask over the feed-forward network intermediate dimensions of the transformer, preserving only the neurons that are important for representing the persona through structured pruning. Experiments on Llama and Qwen instruction-tuned models under different sparsity levels show that the proposed method significantly reduces performance degradation in role-playing tasks compared to existing pruning approaches while effectively compressing the model. The results also indicate that the pruned models maintain relatively stable performance on general reasoning benchmarks.

**Compliance With Llm Reviewing Policy:**

Affirmed.

**Final Justification:**

To reveal the effectiveness of pruning, the gain and the performance should be statistically verified. This is because why I weighed more on soundness than others. The first rebuttal and the second reply from the authors mostly addressed my concerns, except that the statistical difference between LLM-Pruner and theirs may not be statistically different (2-std ranges of those two might overlap). Though I strongly recommend that the authors should not use the term like "outperform" in such cases yet, this paper could provide some insight to the community.

**Key Questions For Authors:**

## Questions on evaluation

1. How can we ensure the quality of automated synthesis (Section 3.1)? For a simple persona as shown in Figure 2, it could be checked automatically. But, what if we use a more complex type of persona, e.g., a character from a fictional work? As the authors targeting role-playing agents and the field already discussing such characters from the work, this should be clarified.

2. With role-playing perspective, the Alpaca set might not provide appropriate query-response pairs, because it mainly asks question about something. Sometimes, role-playing does not query something; rather, role-playing requires appropriate reactions about the situation. Thus, it seems that the current evaluation scheme and the real application on role-playing seems have a gap. Could the authors explain more about this?

2a. Since several applications mentioned in the paper involve conversational agents, it may also be useful to evaluate persona consistency in multi-turn dialogue settings. For example, COSER [1] proposes a framework for evaluating role-playing performance in multi-turn interactions. It would be interesting to see whether the proposed approach maintains persona consistency under such evaluation settings.

[1] Wang, X., Wang, H., Zhang, Y., Yuan, X., Xu, R., Huang, J. T., ... & Xiao, Y. (2025, February). Coser: Coordinating llm-based persona simulation of established roles. In Forty-second International Conference on Machine Learning.

3. Did the persona disjoint across train and test set of Alpaca-P?

4. The authors used small language models as a models to be pruned. What if we use larger size of models? Is the reasoning ability or conversation ability preserved also or harmed? As large models sometimes behave differently from small models, it is better to include results about large models.

5. The paper relies heavily on LLM-as-a-Judge for both dataset construction and evaluation. Given the relatively small dataset size, did the authors consider conducting human evaluation with predefined evaluation criteria to validate role-playing quality? Also, why did the authors used 100-point scoring instead of binary correctness (Section 4, page 5)?

## Questions on the results

6. Though the table presents 50% ratio of pruning, the discussion about 50% is missing. What did the authors think about the result about 50%, compared to 25%? It could be obvious that more pruning leads worse performance, but it seems that general performance seldom drops; only the role-playing performance drops on 50% (without recovery). It seems that the pruning actually eliminates computational procedure related to role-playing, rather than general one. Can the authors explain about this?

7. How prevalent that the erroneous cases which are specified in Section 4.2? Per each model, please give the proportion of each cases, to strengthen the qualitative result.

8. Because of the page limit, I think the authors might have difficulty on explaining "Stability of persona sub-network discovery" and "Structural-semantic alignment". Could you explain more about these?

**Limitations:**

Yes.

**Strengths And Weaknesses:**

## Strength

- The paper is somewhat well written and provides a well-organized overview of related work.
- The research direction is clear and interesting in that it addresses the efficiency problem of role-playing LLMs from a pruning perspective.
- The paper proposes a practical framework that combines persona-based data synthesis with model pruning.
- Experiments show that the method better preserves role-playing performance compared to existing pruning approaches.

## Weakness

- The evaluation is conducted on a very small number of personas (10 in RoleBench and 10 in Alpaca-P), which makes it difficult to confidently attribute performance differences to the proposed method rather than variance in the data. Consequently, the results are unlikely to have statistical significance, and additional experiments with a larger number of personas and samples would be necessary to support the claims more convincingly.
- The paper relies heavily on LLM-as-a-Judge both in the dataset construction and evaluation process. Given the relatively small dataset size, incorporating human evaluation would have been feasible and could provide more reliable validation. Moreover, the query filtering stage uses Llama-3.2-3B as the judge model, while GPT-4o-mini is used for evaluation elsewhere in the paper. This inconsistency reduces the reliability of the filtering process; using the same stronger judge model would likely improve credibility.
- The qualitative evaluation lacks statistical backups. The authors mentioned some cases (generic, degraded responses, or lack of character traits) but the frequency of such cases are missing. To provide a clear view of performance outcomes clearly, it is better to add proportion of such cases and discuss the result based on proportion of each cases.
- The proposed approach appears to require training a separate pruned model for each persona. This raises scalability concerns in practical scenarios with many personas (e.g., large NPC environments), as the number of models would increase proportionally with the number of personas.
- The evaluation mainly focuses on stylistic similarity in generated responses. It is unclear whether the method truly captures deeper persona characteristics or primarily learns superficial stylistic patterns.
- The experiments evaluate role-playing primarily in single-turn settings. However, almost applications mentioned in the introduction (e.g., NPC systems, chatbots simulating user-preferred characters) naturally involve multi-turn interactions. It is therefore unclear whether the proposed method can maintain persona consistency in such multi-turn conversational scenarios.

---

> ### Author Rebuttal · Authors · 2026-03-31
>
> We sincerely thank you for your detailed feedback. We address your major concerns one-by-one below and are happy to provide further clarification if needed. Due to the constraints of the response letter, we provide additional figures and tables through the following (anonymous) link in compliance with the ICML policy: [Supplementary URL](https://anonymous.4open.science/r/anony_fig_tab-8C55)
>
> ---
> **[W1] Statistical significance of evaluation**
>
> Following your suggestion, we conducted a larger-scale evaluation (50 personas × 100 queries). The results are consistent with Table A (Supp. URL), with our method remaining the most effective by a large margin. To further address your concern, we additionally report the average per-sample rank, which highlights a clearer performance gap between our method and the baselines, along with low variance in per-sample rank.
>
> **[W2/W5/Q5] Reliability of LLM-as-a-Judge / Evaluation beyond stylistic similarity**
>
> We additionally conducted a human evaluation to assess the reliability of our LLM judge. Here, 50 participants ranked five model outputs on three criteria in [1]: storyline consistency, anthropomorphism, and character fidelity. As shown in Table B (Supp. URL), human and LLM rankings were highly consistent, with strong Pearson and Spearman correlations (r=0.87, ρ=0.87) and low p-values. In addition, using Qwen3-32B as an additional judge yielded the same conclusion, with our method still outperforming baselines.
>
> **[W3/Q7] Statistical backups for qualitative results**
>
> We manually inspected the failure types of low-quality outputs (persona score below 60) for each model, as reported in Table C (Supp. URL); the baselines tend either to produce generic assistant-like responses, or to generate incomplete responses, infinite repetition, and refusal. We will incorporate these and more analysis in the final draft.
>
> **[W4] Separate training required for each persona?**
>
> Our method does not necessarily require separate training for each persona. Even without recovery finetuning, it can effectively role-play the target persona while preserving general capabilities, as shown in Table 1. That said, performance can be further improved with finetuning, whereas other baselines still struggle to recover performance.
>
> **[W6/Q2a] Multi-turn persona consistency**
>
> We additionally evaluated our method in a multi-turn dialogue setup following [1], with 10-turn conversations between each role-playing model and a simulated partner (GPT-4o). As shown in Table D (Supp. URL), our method outperforms the baselines on all three evaluation dimensions. We also report turn-level persona consistency using the attribute consistency metric [2]. As shown in Figure A (Supp. URL), baseline performance declines over turns, whereas our method remains stable.
>
> **[W2/Q1] Why use Llama instead of GPT-4o-mini for query filtering?; How to ensure the quality of automated synthesis?**
>
> We use Llama because filtering requires access to internal representations, which are typically unavailable in commercial models like GPT-4o-mini. As described in Section 3.1, this filtering scheme can be applied not only to simple personas but also to complex descriptions; for example, the sample in Figure 2 is a shortened illustration of an 8-line persona description, as shown in Table 4 of Appendix B. As shown in Table 5 of Appendix B, leveraging such internal representations enables strong filtering performance even with a relatively small model.
>
> **[Q2] Is the use of Alpaca appropriate for role-playing?**
>
> In our evaluation, the judge is not given any reference response and assesses only role-playing behavior, i.e., the evaluation is independent of the query format. Moreover, we intentionally avoid relying on roleplay-specific datasets, as we aim to develop a scalable method that can extract role-playing data from general-purpose datasets.
>
> **[Q3] Did the persona disjoint across train/test sets?**
> No. We construct one pruned model per persona and use the same persona for training and evaluation. Instead, we split the queries so that evaluation reflects generalization to unseen queries under a fixed persona.
>
> **[Q4] Results on larger models**
>
> Following your suggestion, we additionally tested Qwen-2.5-32B-Instruct. As shown in Table E (Supp. URL), our method effectively retains personalized behaviors, whereas the baselines do not. We note that the 32B model better preserves general capabilities with our method, suggesting that it becomes more effective at larger model scales.
>
> **[Q6] Table 1 (50% sparsity): General performance seldom drops**
>
> Thank you for the insightful observation. Indeed, we consistently observe that persona-oriented pruning is significantly more challenging for maintaining performance than general pruning, which highlights the novelty of our problem setup. We will reflect this point more clearly in the final draft
>
> ---
>
> [1] Wang et al., CoSER. ICML 2025.
>
> [2] Zhou et al., CharacterBench. AAAI 2025.

---

> > ### Author Rebuttal · Reviewer_4brE · 2026-04-04
> >
> > Thanks for the detailed response with additional data. However, the statistical result was not what I expected. Statistical testing is a method that trying to remove errors during statistical or measurements. The result for W1 actually showing the standard deviation of ranks, not the deviation of score. The rank discretizes and emphasizes small differences between models. Therefore, to report a clear and statistically unbiased result, I recommend providing genuine standard deviation instead of ranks.
> >
> > [W2/W5/Q5] Similar to W1. It seems that the result is based on ranks. Then, the correlation should not be measured with Pearson, because Pearson's r assumes that the data follows continuous distribution (rank is not.) Though, it seems that Spearman rank correlation is appropriate and acceptable. It could be better to report raw scores with raw standard deviation, instead of ranks.
> >
> > [W3/Q7] Addressed.
> >
> > [W4] Addressed.
> >
> > [W6/Q2a] If possible, could you show the IQR or standard deviation for Figure A? To avoid over-claiming, I want to re-check the result whether the baselines are actually decreasing (that is, there's no overlap between those two).
> >
> > [W2/Q1] Addressed.
> >
> > [Q2] Addressed.
> >
> > [Q3] When the persona is not disjoint across train/test set, the improvement might seem obvious because the dataset which PersonaPruner was trained on contains valuable information about the persona and the baselines were not trained on them (Correct me if I'm wrong). Are there any other results based on disjoint set of personas?
> >
> > [Q4] Addressed. It might be better to add standard deviation of raw scores.
> >
> > [Q6] Addressed.

---

> > > ### Author Response · Authors · 2026-04-05
> > >
> > > Dear Reviewer 4brE,
> > >
> > > We sincerely appreciate your careful reading of our response and your additional feedback. We provide our responses to the remaining points below, along with a separate Supplementary URL containing an additional figure: [Supplementary URL 2](https://anonymous.4open.science/r/annony_fig_a-A06A)
> > >
> > >
> > > ---
> > >
> > > **AQ1. Standard deviations of raw persona scores [W1, W2/W5/Q5, Q4]**
> > >
> > > Thank you for the suggestion regarding our statistical analysis.
> > >
> > > For LLM-as-judge evaluation, we reported rank-based statistics to provide a normalized summary across methods; since the LLM-as-judge outputs can vary in scale across queries, we compute ranks at the query level to mitigate such variability and ensure consistent comparisons. Nevertheless, we agree that standard deviation on raw scores provides a more direct measure, and we will include it in the final draft. For example, the tables below report the mean and standard deviation of the raw persona scores on Alpaca-P for Llama-3.2-3B-Instruct (25% pruned) under both the original (10P x 20Q) and expanded (50P x 100Q) data scales, as well as for Qwen2.5-32B-Instruct (50% pruned). The updated results show that our method consistently achieves the highest persona scores while maintaining lower variance than the baselines across all settings, except for SliceGPT, which exhibits low variance but significantly lower mean scores, limiting its practical relevance.
> > >
> > > For human evaluation, we adopt a ranking-based protocol to account for the subjective nature of persona consistency, where relative comparisons tend to be more reliable than absolute scoring across annotators. As a result, raw numerical scores are not available under this evaluation protocol. We agree that Pearson correlation may not be appropriate for rank-based data, and will revise the final draft to emphasize Spearman rank correlation as the primary metric.
> > >
> > > | Llama-3.2-3B-Instruct (25%) | 10P × 20Q Alpaca-P (Mean ± Std.) | 50P × 100Q Alpaca-P (Mean ± Std.) |
> > > | --- | --- | --- |
> > > | Depth Pruning | 20.61 ± 18.85 | 21.48 ± 19.18 |
> > > | SliceGPT | 5.32 ± 8.50 | 5.42 ± 10.70 |
> > > | LLM-Pruner | 70.64 ± 15.91 | 61.91 ± 23.21 |
> > > | Adapt-Pruner | 50.74 ± 21.32 | 63.42 ± 23.76 |
> > > | **Persona-Pruner (Ours)** | **80.19 ± 11.65** | **81.97 ± 11.89** |
> > >
> > >
> > > | **Qwen2.5-32B-Instruct (50%)** | **Alpaca-P (Mean ± Std.)** |
> > > | --- | --- |
> > > | Depth Pruning | 20.18 ± 13.80 |
> > > | SliceGPT | 1.09 ± 4.31 |
> > > | LLM-Pruner | 71.01 ± 17.08 |
> > > | Adapt-Pruner | 47.78 ± 20.97 |
> > > | **Persona-Pruner (Ours)** | **83.10 ± 7.46** |
> > >
> > > ---
> > >
> > > **AQ2. Error bars for Figure A [W6/Q2a]**
> > >
> > > Following your suggestion, we have revised Figure A in the original Supplementary URL to include standard deviations as error bars for the turn-level persona consistency scores. The updated result is presented as Figure B in a separate [Supplementary URL 2](https://anonymous.4open.science/r/annony_fig_a-A06A); as shown in the figure, while there is some overlap between methods in the early turns, the baselines exhibit a clear decreasing trend over turns, and the gap between our method and the baselines becomes more pronounced in later turns. We will incorporate this result into the final draft.
> > >
> > > ---
> > >
> > > **AQ3. Results based on disjoint persona sets? [Q3]**
> > >
> > > We note that evaluating generalization across disjoint personas corresponds to a different problem setup. In this work, we propose a new problem of “persona-specific pruning”, where the goal is to adapt a model to a single target persona given only its textual description. This differs from a standard generalization setup over disjoint personas, and instead reflects a practical scenario in which role-playing requires capturing detailed persona characteristics without access to persona-specific dialogue corpora. Here, we propose to synthesize persona-aligned data from a general corpus based on the given persona description and use the data to guide pruning toward the target persona. We will clarify this evaluation setting in the final draft.
> > >
> > > ---
> > >
> > > Thank you again for your continued feedback. We hope these clarifications address your questions.
> > >
> > > Sincerely,
> > >
> > > Authors

---

### Decision · Program_Chairs · 2026-04-30

**Decision:**

Accept (regular)

**Comment:**

This paper received consistent ratings of Weak Accept from all reviewers. The paper proposes Persona-Pruner, a structured pruning framework that extracts persona-specific sub-networks from LLMs for lightweight role-playing. All reviewers acknowledged the novelty of the research direction and the practical motivation. Key concerns included limited evaluation scale and statistical rigor, lack of practical efficiency measurements, and missing comparisons with training-based role-playing methods such as Neeko and LoRA. The authors provided a thorough rebuttal, added multi-turn consistency evaluation, and provided comparisons against Neeko, LoRA, and Humanish-Roleplay. The AC recommends acceptance and encouranges the authors to incorporate the additional experiments.